# GPR56/ADGRG1 is associated with response to antidepressant treatment

Raoul Belzeaux[1,2,3], Victor Gorgievski[4,5], Laura M. Fiori[1], Juan Pablo Lopez[1], Julien Grenier[6], Rixing Lin[1], Corina Nagy[1], El Chérif Ibrahim [2,3], Eduardo Gascon [2], Philippe Courtet[3,7], Stéphane Richard-Devantoy[1], Marcelo Berlim[1], Eduardo Chachamovich[1], Jean-François Théroux[1], Sylvie Dumas[8], Bruno Giros[1], Susan Rotzinger[9], Claudio N. Soares[10,11], Jane A. Foster[9], Naguib Mechawar[1], Gregory G. Tall[12], Eleni T. Tzavara[3,4,5,13], Sidney H. Kennedy[9,10] & Gustavo Turecki [1,13✉]

It remains unclear why many patients with depression do not respond to antidepressant treatment. In three cohorts of individuals with depression and treated with serotonin-norepinephrine reuptake inhibitor ($N = 424$) we show that responders, but not non-responders, display an increase of GPR56 mRNA in the blood. In a small group of subjects we also show that GPR56 is downregulated in the PFC of individuals with depression that died by suicide. In mice, we show that chronic stress-induced Gpr56 downregulation in the blood and prefrontal cortex (PFC), which is accompanied by depression-like behavior, and can be reversed by antidepressant treatment. Gpr56 knockdown in mouse PFC is associated with depressive-like behaviors, executive dysfunction and poor response to antidepressant treatment. GPR56 peptide agonists have antidepressant-like effects and upregulated AKT/GSK3/EIF4 pathways. Our findings uncover a potential role of GPR56 in antidepressant response.

[1] Douglas Mental Health University Institute, Department of Psychiatry, McGill University, Montreal, QC, Canada. [2] Aix-Marseille Univ, AP-HM, CNRS, INT, Inst Neurosci Timone, Hôpital Sainte Marguerite, Pôle de psychiatrie, Marseille, France. [3] Fondation FondaMental, Créteil, France. [4] CNRS (Integrative Neuroscience and Cognition Center, UMR 8002), Paris, France. [5] Université Paris Descartes, Sorbonne Paris Cité, Paris, France. [6] INSERM UMR-S 1124 ERL 3649, Université Paris Descartes, Paris, France. [7] Department of Emergency Psychiatry and Acute Care, Lapeyronie Hospital, CHU Montpellier, Montpellier, France. [8] Oramacell, 75006 Paris, France. [9] Centre for Mental Health, Department of Psychiatry, University Health Network, Krembil Research Institute, University of Toronto, Toronto, ON, Canada. [10] St Michael's Hospital, Li Ka Shing Knowledge Institute, Centre for Depression and Suicide Studies, Toronto, ON, Canada. [11] Department of Psychiatry, Queen's University, Kingston, Ontario, Canada. [12] Department of Pharmacology, University of Michigan, Ann Arbor, MI, USA. [13] These authors jointly supervised this work: Eleni T. Tzavara, Gustavo Turecki. ✉email: gustavo.turecki@mcgill.ca

Major depressive disorder (MDD) is a common psychiatric disorder[1] and one of the leading causes of disability worldwide[2]. Antidepressants are the first-line treatment for moderate to severe major depressive episodes (MDE)[3], and although they are effective, not every patient responds to antidepressant treatment. Approximately 40% of patients respond to their first antidepressant trial, and following multiple trials, response rates increase up to 70%[4]. Antidepressants are thought to act through modulation of mono-amines, but the precise mechanisms whereby they affect therapeutic response, as well as the underlying causes of treatment-response variability, remain poorly understood. Therefore, there is an important need to better understand molecular pathways and mechanisms involved in antidepressant response. In this study, we examined peripheral gene expression in three cohorts of individuals with MDD undergoing antidepressant treatment and identified one gene, G-protein coupled receptor 56 (GPR56) whose expression was consistently associated with antidepressant response. We further characterized the function and signaling properties of this gene in vivo and in vitro, and found it to be related to depressive-like behaviors and executive functioning, and to upregulate classical antidepressant signaling pathways upon activation.

## Results

**Gene expression analysis in the discovery cohort.** Antidepressant response involves a complex interplay between genetic and environmental factors. Using a double-blind, randomized clinical trial design (Fig. 1a), we first set out to investigate mRNA changes associated with antidepressant response in patients undergoing a MDE ($N = 237$) treated with either the antidepressant duloxetine ($N = 112$), a serotonin-norepinephrine reuptake inhibitor (SNRI), or with placebo ($N = 125$), over eight weeks. After treatment, 89 (79.5%) and 51 (40.8%) patients were responders in the duloxetine and placebo arms, respectively (Supplementary Table 1). Using the Human HT-12 v4 Expression Bead Chip (Illumina), we found 42 probes, corresponding to 41 different annotated genes, that were overexpressed following duloxetine treatment in responder patients (FDR < 1%). Two of these probes were also upregulated in the placebo group, but among non-responders, while the remaining 40 were specifically overexpressed in duloxetine responders (Supplementary Table 2). No downregulated probes were found by our initial analysis, while 1752 and 1670 probes were found to be downregulated and overexpressed, respectively, using a $t$-test without correction ($p < 0.05$). G protein-coupled receptor 56 (GPR56), also known as ADGRG1, a G protein-coupled receptor of the adhesion class, was the most significantly upregulated mRNA based on fold change (FC) and q-value during duloxetine treatment (FC = 1.19, $q$-value < 0.01; Fig. 1a). Using a general linear model (GLM) for repeated measures, we confirmed that GPR56 was specifically increased only in patients who responded to duloxetine ($F(1,199) = 8.47$, $p = 0.004$; Fig. 1a). These results were technically validated using qRT-PCR (Supplementary Fig. 1) and were not explained by potential clinical or biological confounders including sex, age, BMI, or sample cellular composition (Supplementary Fig. 2).

**Replication of overexpression of GPR56 during antidepressant response.** We next investigated whether the results observed in our discovery cohort could be replicated in two independent cohorts with similar clinical characteristics, but treated with different antidepressants. The Montréal cohort consisted of patients treated with the selective serotonin reuptake inhibitor (SSRI) citalopram over 8 weeks ($N = 63$, Fig. 1b). Similar to the discovery cohort, we found that peripheral levels of GPR56 mRNA

significantly increased after treatment only in patients who responded to antidepressant treatment ($F(1,61) = 4.273$, $p = 0.043$; Fig. 1b). Our third cohort (Marseille cohort) was designed as a naturalistic 30-week-follow-up study, which investigated patients with depression ($N = 64$) and healthy subjects ($N = 87$) (Fig. 1c). Patients received antidepressant treatment as prescribed by their physician/psychiatrist (Supplementary Table 3). As observed in the other two cohorts, we found that GPR56 mRNA levels significantly increased as a function of response after 8 weeks ($N = 30$, FC = 1.26; paired $t$-test $t = -2.52$, $p = 0.018$) while no change were observed in non-responders ($N = 34$, $t = 0.35$, $p = 0.73$). Interestingly, we also saw no change in untreated healthy control subjects ($t = 0.50$, $p = 0.62$). A GLM for repeated measures confirmed a significant time × group interaction ($F(2,148) = 4.98$, $p = 0.008$; Supplementary Fig. 3). Moreover, we found that GPR56 mRNA remained stably overexpressed over a 30-week-follow-up among those who initially responded and then achieved remission after 30 weeks of treatment (responders-remitters, $N = 20$) in comparison to others ($N = 44$) (linear mixed model including 0 week, 2 week, 8 week and 30-week-follow-up, $F(1,230.199) = 14.79$, $p < 0.001$; Fig. 1c).

**Effects of unpredictable chronic mild stress (UCMS) on Gpr56.** Our results above suggest that GPR56 may be involved in mechanisms associated with antidepressant response that are common to different classes of antidepressants, but interestingly, not involved in mechanisms of placebo response. To further examine the potential function and regulation of Gpr56 in depression and antidepressant response, we conducted studies in animals, using the unpredictable chronic mild stress (UCMS) paradigm, a well validated murine model of depression[5], followed by treatment with fluoxetine, a standard SSRI, to model antidepressant effects (Fig. 2a). In this model, stress-exposure leads to depressive-like behaviors (displayed as increased anhedonia and/or resignation) that can be alleviated by subsequent administration of an antidepressant. We have previously adapted this model to distinguish between responder and non-responder mice[5]. Here, chronic stress-induced depressive-like behaviors were effectively reversed in 60% (responders) of the fluoxetine-treated mice. Thus, we investigated peripheral Gpr56 mRNA levels in mice subjected to UCMS and found a significant decrease in mice that manifested depressive-like symptoms as compared to non-stressed mice (FC = 0.81; Fig. 2b). Interestingly, reversal of the depressive-like behaviors with antidepressant treatment was paralleled by normalization of blood Gpr56 mRNA expression in responder mice, i.e., demonstrating improvement in depressive-like phenotype. In contrast, blood Gpr56 mRNA levels remained low in non-responder mice, in close analogy to the Gpr56 expression biosignature seen in the human studies detailed above ($F = 6.15$, $p = 0.001$; Fig. 2b).

Using the same model, we then sought to investigate the effects of stress-induced depression and antidepressant treatment on Gpr56 expression in the central nervous system (CNS). We focused on four regions of interest: the dorsal and ventral hippocampal areas (HD and HV, respectively), the prefrontal cortex (PFC) and the Nucleus Accumbens (NAcc), all previously implicated in stress and depression, albeit in a different manner[6]. A repeated measures two-way ANOVA analysis between groups and brain regions showed a significant interaction between brain region and phenotypes ($F(9,151) = 3.112$; $p = 0.0018$) (Fig. 2c). Subsequent post hoc analyses showed that stress-induced a significant reduction in Gpr56 expression in the PFC (FC = 0.65, $p = 0.0006$) and no effect in the NAcc ($p = 0.08$). In the hippocampus we observed a difference in the HD (FC = 0.62, $p = 0.0001$), in accordance with a previous study[7], but not in the

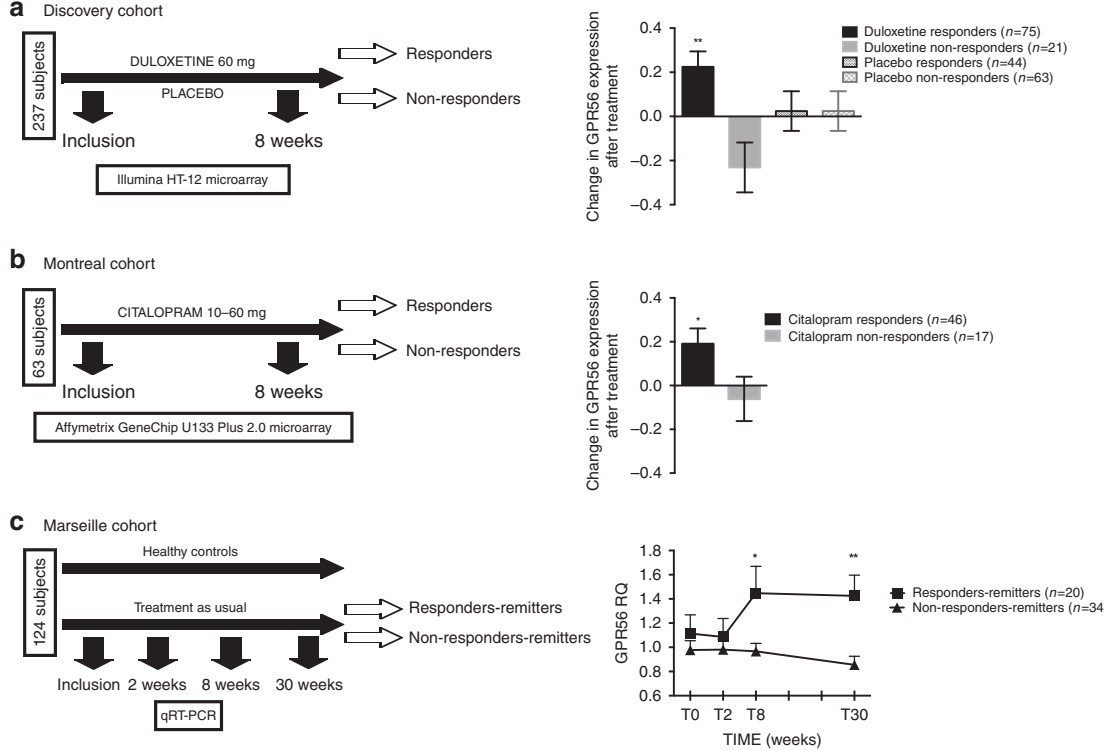

**Fig. 1 GPR56 mRNA is related to antidepressant response. a** In the discovery cohort, 237 patients in a major depressive episode were randomized to double-blind treatment with either duloxetine ($n = 112$) or placebo ($n = 125$), for up to 8 weeks. Using two-class paired significant analysis of microarray (SAM) with correction for multiple testing (FDR < 1%) in patients who responded to duloxetine, *GPR56* mRNA is the most significantly upregulated mRNA in whole blood after duloxetine treatment, based on fold change and *q*-value (FC = 1.19, *q*-value < 0.01). General linear model (GLM) demonstrated a time x treatment interaction, $F_{(1,199)} = 8.468$, $p = 0.004$, that confirms the specificity of *GPR56* mRNA increase in responders to duloxetine. **b** In the first replication cohort (Montréal), patients were treated with citalopram in an open-label trial. *GPR56* mRNA in whole blood demonstrated an increase only in responders. GLM demonstrated a time x group interaction, $F_{(1,61)} = 4.27$, $p = 0.043$ (not adjusted for multiple testing). **c** In the second replication cohort (Marseille), psychiatrically healthy subjects and patients with depression were included in a naturalistic design. In patients who responded and achieved remission after 30 weeks of treatment (responders-remitters, $n = 20$), *GPR56* mRNA is not different at inclusion and 2 weeks, however was then overexpressed at 8 weeks in comparison to others ($n = 44$) (two-sided *t*-test $t = 2.085$, $p = 0.049$) and remained stably overexpressed over a 30-week follow-up ($n = 18$ responders-remitters in comparison to others $n = 31$) two-sided *t*-test $t = 3.076$, $p = 0.005$); Linear Mixed model ($F_{(1,230.199)} = 14.79$, $p = 0.0001$). Bars represent mean. Error bars represent standard error of the mean. **$p < 0.01$, *$p < 0.05$. Source data are provided as a Source Data file.

HV ($p = 0.88$). In the PFC, we observed a bimodal regulation of *Gpr56* mRNA by UCMS and antidepressant treatment. UCMS exposure led to reduced *Gpr56* mRNA expression (Fig. 2c), which was normalized by antidepressant administration in responder mice, but not in non-responder mice (Fig. 2b), a pattern remarkably similar to that seen in the mouse and human blood samples. In contrast, in the HD, while UCMS induced a downregulation of *Gpr56* mRNA, antidepressant treatment had no effect in responder mice ($p = 0.38$). Although brain-blood correlation of gene expression remains a matter of debate, it is worth noting that *Gpr56* mRNA levels were significantly correlated between the PFC and peripheral blood in stressed mice ($r = 0.51$; $p = 0.02$; Supplementary Fig. 4) while in control mice, we found no correlation between blood and PFC ($p = 0.21$). Taken together, our results suggest that an increase in *Gpr56* expression levels may be an integral part of effective antidepressant action. Our results also suggest a specific role for the PFC in relation to *Gpr56* mRNA variation in depressive-like behaviors and antidepressant action, as we found no significant effect of antidepressant-related regulation of *Gpr56* in several other brain regions, including the HD.

**Effects of Gpr56 over-expression and knockdown on mouse behavior.** To investigate a possible causal relationship between *Gpr56* mRNA variation in the PFC and behavioral responses to

stress, we used a viral vector strategy to locally manipulate *Gpr56* expression levels selectively in the PFC, and thus determine the influence of increased or decreased expression of *Gpr56* on depressive-like behaviors and/or antidepressant action in the mouse (Fig. 3 and Supplementary Fig. 5). In naive mice, bilateral PFC infusions of a lentivirus-*Gpr56* construct resulted in PFC *Gpr56* overexpression (FC = 2.02, $t = 4.09$, $p = 0.003$, Supplementary Fig. 5B), while bilateral PFC infusions of a lentivirus-sh-*Gpr56* construct resulted in PFC *Gpr56* downregulation (FC = 0.49, $t = 3.37$, $p = 0.007$, Supplementary Fig. 5C). Behavioral analysis showed that PFC *Gpr56* downregulation was sufficient to produce depressive-like behaviors in unstressed mice, as seen by increased immobility in the TST (Fig. 3b, $t = 2.2$, $p = 0.048$). This test is among the most commonly used procedures to detect clinically effective antidepressant agents because of its high degree of predictive validity, and has been previously used to identify mouse strains that are resistant or hyporesponsive to treatment[8]. In the same test, PFC *Gpr56* overexpression in naive mice induced the opposite effect, namely decreased immobility, the hallmark effect of antidepressant action (Fig. 3c, $t = 3.07$, $p = 0.005$). In both cases there was no effect on locomotor activity, indicating no change in general ambulatory behavior (Supplementary Figs. 6A and 7A), but rather a targeted effect of *Gpr56* PFC manipulations on stress-triggered behaviors. This observation was further strengthened by similar results in the forced

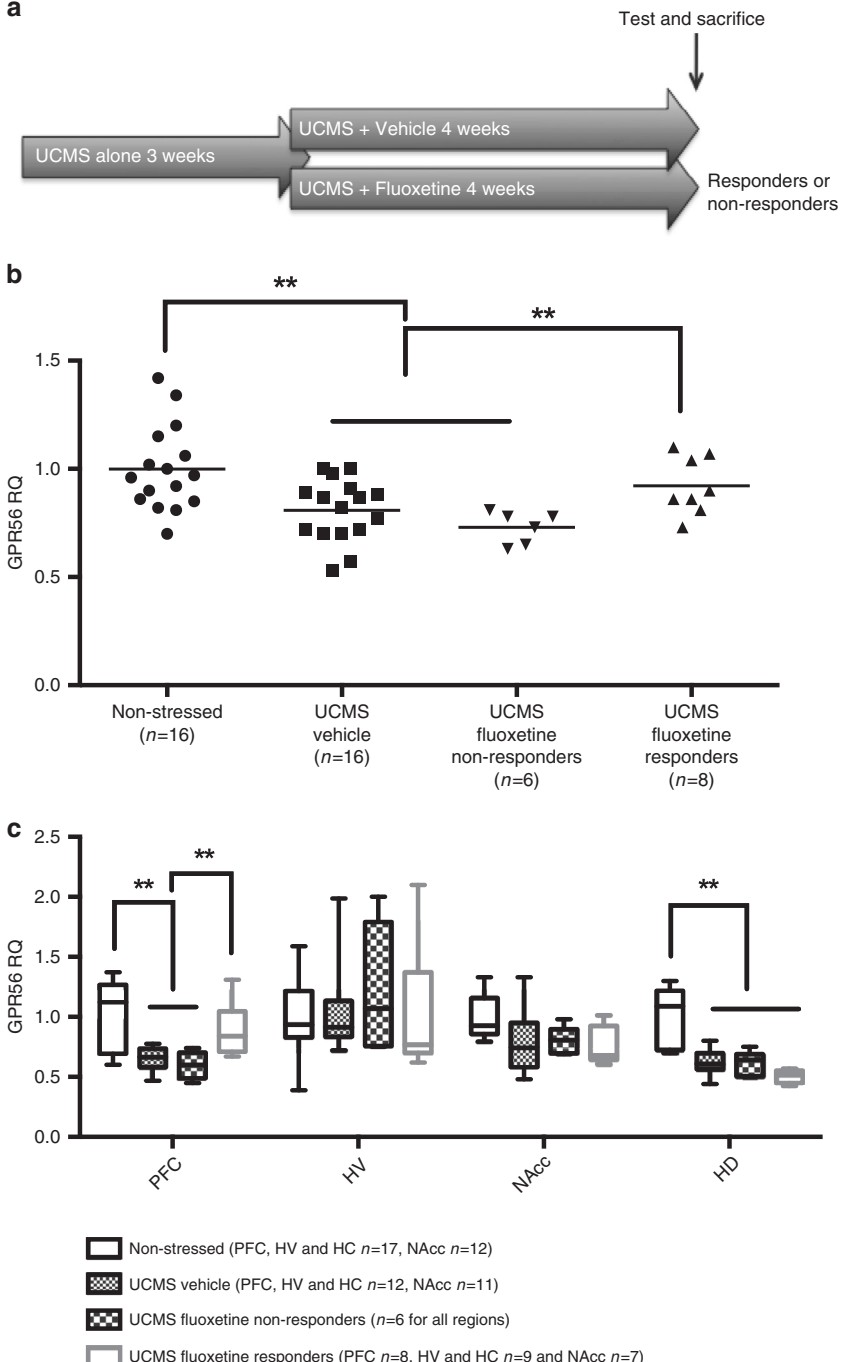

**Fig. 2 Unpredictable chronic mild stress (UCMS) and antidepressant response dysregulate *Gpr56* mRNA in blood and CNS in mice. a** *Gpr56* expression was analysed by qRT-PCR in blood and several brain regions, including the dorsal and ventral hippocampal areas (HD and HV, respectively), the prefrontal cortex (PFC) and the Nucleus Accumbens (NAcc) in non-stressed or stressed mice (exposed to UCMS) and receiving vehicle or fluoxetine. Mice treated by fluoxetine were classified as "responders" or "non-responders" according to behavioral tests. **b** In whole blood, a one-way ANOVA showed between group differences for *Gpr56* expression ($F = 6{,}150$, $p = 0.001$). Blood *Gpr56* mRNA expression was decreased in mice subjected to UCMS, while reversal of depressive-like behaviors with fluoxetine was paralleled by normalization of blood *Gpr56* mRNA expression in responder mice (post hoc analysis $p < 0.01$). **c** In brain, a two-way ANOVA between group and brain regions showed a significant interaction between brain region and mice group ($F(9{,}151) = 3.112$; $p = 0.0018$). Post hoc analysis demonstrated a specific PFC effect, a decrease of *Gpr56* in PFC between stressed and non-stressed mice, with a reversal effect of antidepressant only in responder mice. Sample numbers vary between tissues due to removal of poor quality RNA samples from the analyses. Bars represent mean. Error bars represent standard error of the mean. **$p < 0.01$. Graph represents Box and Whiskers Min to Max. Source data are provided as a Source Data file.

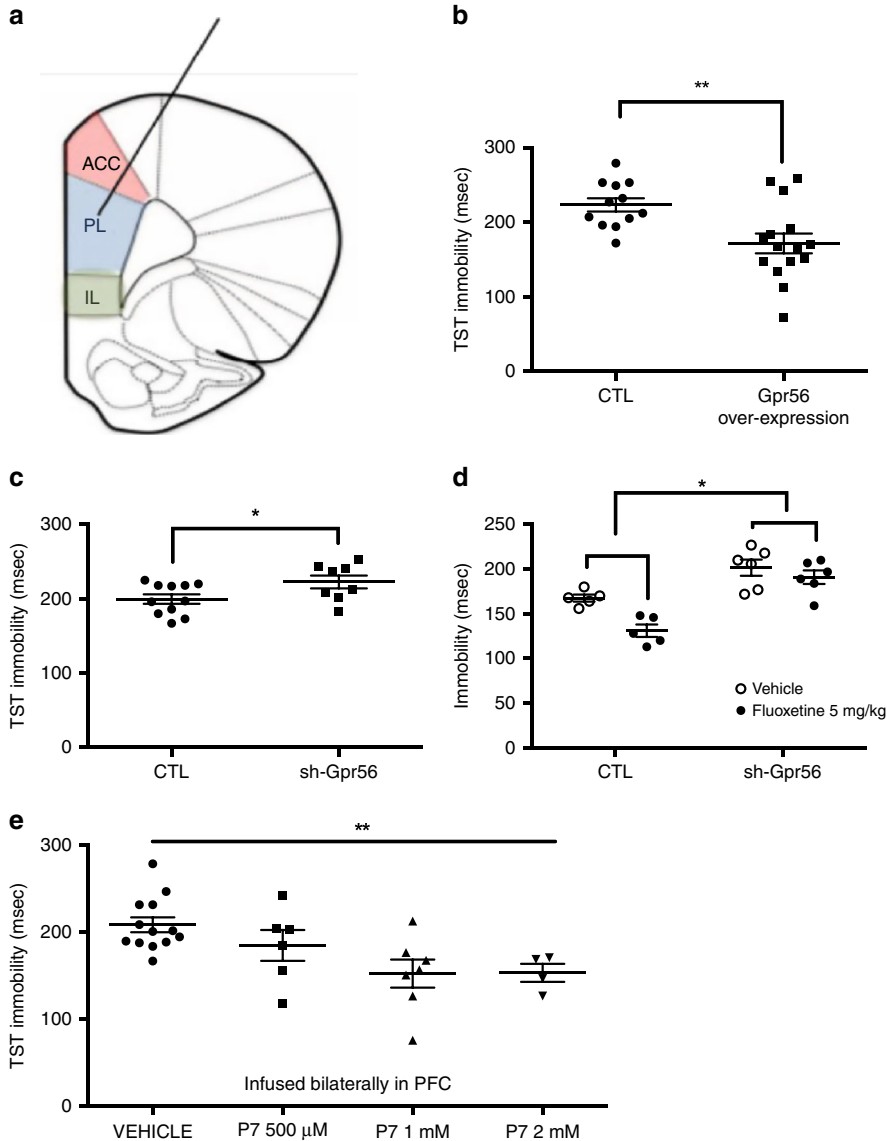

**Fig. 3 Gpr56 regulates depressive-like behaviors. a** Mice were injected bilaterally in the pre-frontal cortex with control (CTL) or lentivirus-*Gpr56* constructs, as well as with CTL or lentivirus-sh-*Gpr56* (inhibitor) constructs. **b** Overexpression of *Gpr56* ($n = 15$) was associated with lower immobility time in the TST in comparison to control animals ($n = 12$), two-sided $t = 3.07$, $p = 0.005$. **c** Downregulation of *Gpr56* ($n = 8$) produced increased immobility time, i.e. depressive-like behavior in the tail suspension test (TST) in comparison to control animals ($n = 11$), two-sided $t = 2.203$, $p = 0.048$. **d** Fluoxetine decreased immobility in control animals ($n = 5$), while this effect was strongly attenuated in sh-*Gpr56*-virus infused animals ($n = 6$); two-way ANOVA for repeated measures; treatment × group interaction $F(1,9) = 6.80$, $p = 0.028$; main effects for group $F(1,9) = 25.4$, $p = 0.001$ (as compared to control) and for treatment $F(1,9) = 6.80$ $p = 0.001$ (as compared to vehicle). **e** GPR56 agonist P7 has antidepressant-like effects. GPR56 agonist P7 peptide infused bilaterally with escalating doses in PFC decreases immobility time and demonstrates antidepressant-like effects, ANOVA $F(1,4) = 4.88$ $p = 0.008$. In comparison to vehicle ($n = 13$), both 1 mM ($n = 7$) and 2 mM ($n = 4$) doses demonstrate a significant decrease of immobility time ($p < 0.01$ and $p = 0.01$ respectively, two-sided post hoc test). Bars represent mean. Error bars represent standard error of the mean. **$p < 0.01$, *$p < 0.05$. Source data are provided as a Source Data file.

swim test (FST) for both up- and downregulation of Gpr56 (Supplementary Figs. 6B and Fig. 7B). We also found behavioral effects for *Gpr56* downregulation in the sucrose preference and O-maze tests (Supplementary Fig. 7C–D). Cognitive symptoms are often associated with MDE and may have prognostic and therapeutic implications, in particular related to executive functioning and PFC functioning[9]. As a consequence, we also conducted a set shifting test (SST), a well validated cognitive test in mice related to PFC functioning, in sh-*Gpr56*-PFC mice[10]. Interestingly, in these mice PFC specific *Gpr56* downregulation is accompanied by impairments in the SST (Supplementary Fig. 8). Overall, these results indicate that *Gpr56* downregulation in the

PFC of non-stressed mice produces depressive-like responses similar to those induced by UCMS. On the contrary, *Gpr56* overexpression in the PFC of naive mice produces behaviors similar to those elicited by classical antidepressants in the TST or FST.

In order to directly probe the link between increased PFC *Gpr56* mRNA and antidepressant response, mice underexpressing *Gpr56* were tested for their response to fluoxetine in the TST. Mice infused in the PFC with control virus or with sh-*Gpr56*-virus were injected acutely with saline or fluoxetine (5 mg/kg) 30 min before the TST trial. Fluoxetine decreased immobility in control animals, while this effect was strongly attenuated in

sh-*Gpr56*-virus infused animals (treatment*group interaction $F$(1,9) = 6.80, $p$ = 0.028; Fig. 3d).

Overall, we showed opposite and bidirectional associations between *Gpr56* expression in the PFC, and depressive- and antidepressant- like responses. Indeed, chronic stress decreased *Gpr56* expression in the PFC whereas antidepressant response normalized this downregulation. *Gpr56* overexpression in the PFC produced antidepressant-like effects, whereas downregulated *Gpr56* in the PFC produced depressive-like behavior, executive function alterations and impaired antidepressant response. Thus, our animal-model results suggest that *Gpr56* may have an important role in the adaptations to stress or in depressive-like behaviors and antidepressant response, and that the PFC is a key region involved in these effects.

**Effects of Gpr56 agonist treatment on mouse behavior.** Following activation of the GPR56 receptor by its ligands, the extracellular and transmembrane domains of GPR56 dissociate to reveal a tethered-peptide-agonist[11]. Based on this mechanism, synthetic peptides (i.e., P7 "TYFAVLM-NH2" and P19 "TYFAVLMQLSPALVPAELL-NH2"), comprising the specific portion of the tethered-peptide-agonist, have been generated and demonstrate GPR56 agonist properties[11,12]. We bilaterally infused these peptides and their inactive controls in the mouse PFC to explore the behavioral effects of GPR56 activation. Parallel to our results with Gpr56 overexpression, behavioral analyses showed that GPR56 agonists produced antidepressant-like effects in unstressed mice, as seen by decreased immobility in the tail suspension test (TST) for P7 with a dose-response profile (Fig. 3e, ANOVA $F(1,4) = 4.88$ $p = 0.008$) and for P19 (Supplementary Fig. 9). Interestingly, we confirmed the specificity of antidepressant-like effects of GPR56 in the PFC by using the same peptides infused in the NAcc, which produced no behavioral effects (Supplementary Fig. 10). Finally, the antidepressant-like effect of the GPR56 agonists was not explained by basic locomotion differences, as we did not find any differences in ambulations across time between active peptides and their controls (Supplementary Fig. 11). These data provide evidence that activation of GPR56 through pharmacological manipulation by GPR56-specific ligands has antidepressant-like-effects, specifically in the PFC. These experiments further support a role of GPR56 in depressive-like behaviors and antidepressant response. Furthermore, they indicate that GPR56 may represent a molecular target for treatment of MDD.

**GPR56 expression in depressed human brains and relation to executive function.** Similar to our results in mice (Fig. 2), *GPR56* is expressed in all brain regions in humans (Supplementary Fig. 12). Therefore, we next investigated the expression of *GPR56* in the PFC (BA44) from individuals who died during an episode of MDD and compared them with psychiatrically healthy individuals. We found that *GPR56* expression was significantly lower in cases in comparison to controls (FC = 0.56, $U = 385$, $Z = -2.81$, $p = 0.005$, Fig. 4a). Controlling for covariates and possible confounders, such as age, sex, PMI and tissue pH, did not have an impact on our results ($F(1,69) = 4.91$, $p = 0.030$).

A central role of the PFC is executive function. Thus, we hypothesized that variation of *GPR56* levels associated with antidepressant response may result in cognitive changes in patients, a hypothesis consistent with previous data suggesting that improvement in executive functioning is associated with antidepressant response[9]. Neuropsychological testing was conducted in a subset of individuals who participated in the duloxetine trial. We investigated executive functioning using the Stroop interference test, and analyzed the data as a function of *GPR56* mRNA variation. Variation in the Stroop interference score was negatively correlated with variation of *GPR56* mRNA levels ($r = -0.71$, $p = 0.009$, Fig. 4b), associating increased *GPR56* mRNA with improved executive functions, a finding that mirrors results in the mouse. A partial correlation analysis confirmed this result after correction by response status, age, and gender (correlation = $-0.796$, df = 7, $p = 0.01$).

**GPR56 agonists upregulate AKT/GSK3/EIF4 pathways in neuroblastoma cells.** In order to gain a more complete understanding of downstream signaling processes initiated by GPR56 activation, we investigated the transcriptional consequences of treatment with the two GPR56 agonists described above. These experiments were performed in vitro, using a human neuroblastoma cell line, which was treated with the agonist peptides for 24 h then examined using RNA sequencing. We used these cells because they are derived from neural cells, express GPR56 receptors, and express genes from several important pathways that have been associated with antidepressant response, including the serotonin signaling pathway[13,14].

To functionally characterize the gene expression variation associated with agonist-induced activation of GPR56, we used Gene Set Enrichment Analysis[15]. 6,568 gene sets with sizes between 15 and 500 genes were included in the analysis after gene set size filtering. Among them, we identified significant enrichment of nine gene sets (FWER < 0.20, Supplementary Table 4). Interestingly, AKT, GSK3 and EIF4 pathways demonstrated the highest normalized enrichment scores and lowest FWER *p*-values for upregulated gene sets. These pathways were upregulated in cells treated with the agonists, in comparison to control conditions. These pathways are highly related and have been described as downstream biological mechanisms involved in depression and antidepressant action of several different drugs, including SSRIs and ketamine[16–19]. As a consequence, our results suggest that GPR56 agonists may have antidepressant effects through pathways that are similar to those activated by established antidepressants.

**Discussion**

Taken together, our preclinical and clinical results identified GPR56 as a player in depressive symptomatology and a key mediator of antidepressant response in blood and in the brain. In blood, *GPR56* mRNA increased in parallel to antidepressant response and could be used to monitor antidepressant response. In the brain, namely in the PFC, decreased *GPR56* expression associated with depression in humans or depressive-like behaviors in mice, whereas in mice increased PFC *GPR56* expression was necessary and sufficient for antidepressant action, an effect that might involve cognitive modulation. Using two agonist peptides, we confirmed in mice that activation of GPR56 in the PFC is associated with behavioral responses that are commonly associated with antidepressant treatment. Moreover, based on cell experiments and RNA sequencing, we found that GPR56 agonists upregulated AKT-GSK3-EIF4 pathways, downstream biological mechanisms previously associated with depression and antidepressants action[16–19]. Although we did not examine these pathways in vivo, it is possible that the upregulation of these pathways explains the antidepressant effects that were observed following agonist treatment. Overall, our results suggest that GPR56 is a potential target for development of antidepressant drugs.

GPR56 is involved in a number of biological functions relevant to the pathophysiology of depression, including neurogenesis, oligodendrocyte development and progenitor cell migration in brain, as well as myelin repair[20–23], in parallel to its important role in

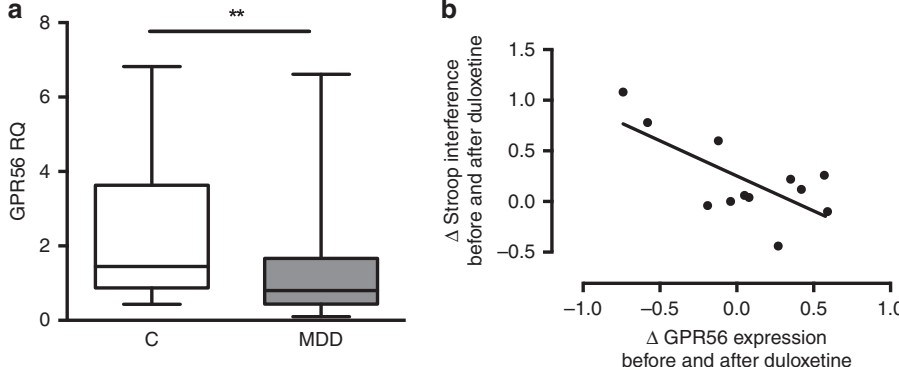

**Fig. 4 GPR56 expression is altered in the prefrontal cortex from post-mortem brain tissue of individuals with depression and is related to pre-frontal cortex functioning in patients. a** GPR56 expression was measured by qRT-PCR in post-mortem brain tissue (BA44). Expression was lower in individuals with depression (MDD, $n = 49$) in comparison to psychiatrically healthy controls ($n = 26$). FC = 0.56, two-sided $U = 385$, $Z = -2.81$, $p = 0.005$. Graph represents Box and Whiskers Min to Max. **b** Changes in Stroop interference score, a neuropsychological test that involves pre-frontal cortex function, were correlated with changes in GPR56 expression in whole blood following antidepressant treatment ($n = 12$ from discovery cohort, Pearson coefficient of correlation = $-0.71$, two-sided $p = 0.009$). Reduction of interference score was associated with an improvement of Stroop interference test, i.e., an improvement in pre-frontal cortex functioning. Source data are provided as a Source Data file.

immune cell functioning[24–26]. GPR56 ligands comprise two general subtypes: (1) proteins from the surface of neighboring cells, and (2) extracellular matrix proteins. Known extracellular ligands of GPR56 include collagen III, transglumatinase 2, and heparin[23,27,28]. Following activation of the GPR56 receptor by its ligands, the extracellular and transmembrane domains dissociate to reveal a tethered-peptide-agonist[11]. To date, it remains unclear which ligand could be related to depressive behavior and antidepressant effects of GPR56 in the PFC. GPCRs are particularly appealing drug targets. While further efforts are needed to develop compounds with optimised pharmacokinetic properties for in vivo administration, our findings identify a previously unsuspected GPCR as a possible target for antidepressants.

Our study does have several limitations. Firstly, the human cohorts have relatively modest sample sizes that may limit the generalisation of our results. However, we replicated our findings in three relatively different treatment cohorts. Secondly, as we only conducted these studies using SSRI and SNRI antidepressants, we cannot conclude if GPR56 is involved in antidepressant response in general or limited to specific antidepressant classes. Thirdly, GPR56 is expressed in numerous cell types and tissues, and is involved in different processes that may be related to antidepressant response[21–25]. In the brain, single-cell sequencing data from the frontal cortex indicates that GPR56 is expressed in all cell types, including glutamatergic neurons, and astrocytes, where relatively higher levels are observed[29]. These data are consistent, although not fully concordant, with results we observed in the mouse using fluorescence in situ hybridization (Supplementary Fig. 13). In the future, cell-type specific studies should be conducted in order to elucidate the mechanisms through which GPR56 is involved in depression and antidepressant response. It remains unclear how GPR56 may be regulated by antidepressants in both the brain and blood. Moreover, it is unclear if variation in the blood only mirrors a process in the brain, or reflects an active biological process that influences antidepressant response. Fourthly, we conducted our animal experiments to test the causal relationship between Gpr56 under- and over-expression, as well as pharmacological testing, only in acute stress paradigms of depressive-like symptoms (i.e., TST, FST).

Despite these limitations, by integrating several clinical cohorts, animal studies and post-mortem brain analyses, our results provide a greater understanding of the pathophysiology of depression, and suggest a drug target for the treatment of MDD.

## Methods

**Study design**. *Participants*: Our study involved three cohorts of living subjects and one cohort of post-mortem brain samples. Informed written consent was obtained from all participants, or next-of-kin. Each study was approved by the appropriate ethics committee.

The discovery cohort consisted of 237 patients in a MDE (69.6% female) who were randomized to double-blind treatment with either duloxetine (60 mg die, $N = 112$), a SNRI, or placebo ($N = 125$), for up to 8 weeks (www.ClinicalTrials.gov 11984A NCT00635219; 11918A NCT00599911; 13267A NCT01140906). These studies were sponsored by Lundbeck and samples were provided as a donation to the Canadian Biomarker Integration Network in Depression program. Patients were excluded from the study if they suffered from bipolar disorder, schizophrenia and/or comorbid substance use disorder. Depressive symptoms were assessed using the Montgomery Åsberg Depression Rating Scale (MADRS) at baseline and at the end of the trial. The mean age was 46.8 years ± 12.8 SD. Before treatment, mean depression severity level using MADRS total score was 31.2 ± 3.7 SD, ranging from 26 to 46. Patients were classified as responders or non-responders according to a MADRS reduction of ≥50% or <50%, respectively. Haematological data (blood cell counts) were collected at inclusion and at the end of treatment.

The first replication cohort (Montréal cohort) was an independent group of 63 patients treated with citalopram, a SSRI, in an open-label trial previously published[30]. This study was conducted in Douglas Mental Health Institute with the necessary approval from the hospital ethics and internal review board. In this study, patients experiencing a MDE received citalopram for 8 weeks. The same exclusion criteria as above were applied. At the end of the follow-up, patients were classified as responders ($N = 46$) or non-responders ($N = 17$), according to a Hamilton Depression Rating Scale 21 items (HDRS-21) reduction of ≥50% or <50%, respectively.

Our second replication cohort (Marseille cohort) was a naturalistic prospective cohort that included 64 patients and 87 healthy controls (ClinicalTrials.gov NCT02209142)[31]. Patients were included during a MDE with HDRS-17 > 19 at the inclusion. Patients were excluded from this study if they suffered from bipolar disorder, schizophrenia and/or comorbid substance use disorder. All patients were treated at the inclusion with treatment as usual upon discretion of the treating psychiatrist (Supplementary Table 3). Healthy controls were free of any psychiatric disorder according to a semi-structured interview. All subjects included in the analysis were followed for 30 weeks with four points of evaluation (i.e., inclusion, 2 weeks later, 8 weeks later and 30 weeks later). After an 8-week-follow-up, 30 patients were classified as responders according to a HDRS-17 reduction of ≥50%. At the end of the study (30 weeks), 20 patients were classified as responders at 8 weeks that achieved remission at 30 weeks (i.e., HDRS-17 ≤ 7) and 44 patients were classified as never responders or responders without remission.

The post-mortem cohort was comprised of 75 post-mortem PFC samples (Brodmann Area 44) obtained from the Douglas-Bell Canada Brain Bank (Douglas Mental Health University Institute, Montreal, Quebec, Canada). Ethics approval was obtained from the institutional review board of the Douglas Mental Health University Institute. Brain pH and post-mortem interval (PMI) were used as tissue integrity measures. Subjects were either individuals who were suffering from a MDE at time of death by suicide ($N = 49$), or psychiatrically normal controls ($N = 26$), as assessed by psychological autopsies using DSM-IV critepost-mortem intervalria.

**Sample collection and processing**. Whole blood samples were collected using PAXgene Blood RNA Tubes (PreAnalytix®) for the discovery cohort and Montréal cohort, while Marseille samples were collected in EDTA tubes and later processed using Leukolock filters. Brain tissues (post-mortem cohort) were processed and dissected at 4 °C, then snap-frozen in liquid nitrogen before storage at −80 °C. Total RNA was extracted using the Qiagen miRNeasy Micro Kit (discovery, Montreal, post-mortem cohorts) or Ambion spin columns (Marseille cohort), with DNase treatment. RNA integrity was evaluated using an Agilent Bioanalyzer. All samples had a RNA integrity number (RIN) > 6.

**Microarray quantification and data processing**. RNA from the discovery cohort was hybridized to the Illumina Human-HT-12 v4 microarray. Samples were randomized to avoid batch effects. All array probes and samples were subjected to quality control using Flexarray® package implemented in R (version 1.6.3). Data were normalized using background adjustment and log2 transformation, variance stabilization transformation (VST) correction, and quantile normalization. A principal component analysis was used to identify outliers resulting in identification and exclusion of 13 samples. After exclusion of outliers, 443 remaining samples were re-used in the same normalization procedure, comprising 237 different individual subjects. In total, 47,323 probes were present in the microarray. All probes were filtered using a detection $P$ value < 0.01 in at least 10% of the samples, resulting in available expression data for 16,674 remaining probes.

Gene expression from the Montréal cohort was measured in whole blood using the Affymetrix GeneChip Human Genome U133 Plus 2.0 array as previously described[30]. We used normalized and filtered data to assess the expression of GPR56 in this data set.

**Quantitative real-time polymerase chain reaction (qRT-PCR)**. For the discovery and post-mortem cohorts, total RNA was reverse-transcribed using M-MLV Reverse Transcriptase (200 U/uL) (ThermoFisher®) and oligo (dT) 16 primers (Invitrogen). Real-time PCR (qRT-PCR) were run in triplicate using the Quant-Studio™ 6 Flex System and data collected using QuantStudio™ Real-Time PCR Software v1.1. Expression levels were calculated using the absolute (standard curve method) or relative ($2^{-\Delta\Delta Ct}$) quantification method, depending on experimental design. GAPDH was used as the endogenous control. The following primers were used in the study: GPR56 (FW: CCCATCTTTCTGGTGACGCT; REV: GATCC AGCACATGGAAGGGT) and GAPDH (FW: TTGTCAAGCTCATTTCCTGG; REV: TGTGAGGAGGGGAGATTCAG).

For the Marseille cohort, total RNA was reverse transcribed using the High-Capacity cDNA Reverse Transcription kit (Life Technologies, Applied Biosystems, Foster City, CA). Real-time PCR reactions were performed in duplicate using the TaqMan Universal PCR Master Mix II with no UNG (Life Technologies, Applied Biosystems, Foster City, CA), with an ABI PRISM 7900HT thermocycler under the following conditions: 10 min at 95 °C, 50 cycles of 15 s at 95 °C and 1 min at 60 °C. Primers/TaqMan probe assays (Hs00173754_m1) purchased from Applied Biosystems were used to determine the level of expression of GPR56 transcripts. We used CRYL1 (Hs00211084_m1) as a reference gene as previously described[32]. Expression levels were calculated using the relative ($2^{-\Delta\Delta Ct}$) quantification method.

**Evaluation of executive functioning**. For a sub-sample of the discovery cohort, executive functioning was evaluated using a standard color-word Stroop task at inclusion and after treatment ($N = 12$). In this task, the participant is asked to name the colors of a series of words "red," "green," and "blue" as quickly as possible without making mistakes. During a congruent task, the actual observed colors of the words match the colors that the words denote, while during an incongruent task, the series of color words does not match with the actual color. The interference Stroop score was calculated as the difference between time of reading during incongruent and congruent tasks. As such, the higher the interference score is, the greater the impairment of executive functioning.

**Mouse studies**. *Animals*: All mice used were male adults (3–6-months old). The following strains were used: C57Bl6, and BALB/cJico. All UCMS experiments were performed using BALB/c mice, as previously published[33]. Targeted manipulations of Gpr56 expression in the PFC through lentiviral particle infusions were performed using C57Bl6. The mice were kept under standard conditions at 22 ± 1 °C, and a 12-h light-dark cycle with food and water available ad libitum except when food/water deprivation was part of the experimental protocol. Humidity levels were between 45 and 55%. Behavioral assessments were performed during the second half of the light phase. All animal protocols and welfare complied with French and European Ethical regulations. The experimental protocols were approved by the local Ethical Committee (Comité d'éthique en expérimentation animale Charles Darwin N°5).

Unpredictable chronic mild stress (UCMS): After a two week acclimation period BALB/c male mice (8-week old) were individually housed and subjected to UCMS as described[33]. Stressors, typically wet bedding, tilted cages, lights on at night, crowding, difficult access to food, paired housing with intruder restraint and forced swim were applied twice a day for a two hour period and overnight in a randomized order. No food deprivation was used. Control (non-stressed) mice were standard housed in a room distinct to that of the stressed mice. Throughout

the UCMS protocol the animal's weight was measured every five days. At the end of the chronic stress protocol the emotional state of the animals was evaluated in the TST and the sucrose preference tests as described[33]. The stress procedure was maintained with items compatible with behavioral testing. Control (non-stressed) mice were left undisturbed throughout the protocol.

Pharmacological effects and response to antidepressants in UCMS-subjected mice were assessed with a reversal protocol (45 days of UCMS; treatment during the last 3 weeks) as previously described[5]. Namely, mice were subjected to the UCMS-protocol for 45 days starting Day 0. During the first 3 weeks there was no treatment for any of the groups. From week 4 until the end of the protocol, mice were treated daily with saline or the reference antidepressant fluoxetine at 15 mg/kg, i.p., daily. In the literature it has been reported that fluoxetine administered in a reversal mode exerts a bimodal effect; it elicits a response in a sub-group of responder mice and has no effect on a distinct sub-group of non-responders[34]. We therefore sought to distinguish between fluoxetine responders and non-responders for subsequent qPCR determinations of Gpr56 mRNA expression.

*Lentiviral particle-infusions in the PFC*: Lentiviral particles: The following commercially available lentiviral particle solutions were used: for Gpr56 overexpression, we used the Adgrg1 (NM_018882) Mouse Tagged ORF Clone Lentiviral Particle, >$10^7$ TU/mL (Origene CAT#: MR210044L2V), and for Gpr56 downregulation, we used Gpr56 shRNA (m) Lentiviral Particles (Santa Cruz: sc-60750-V).

Infusions: For stereotaxic delivery, mice were anesthetized with a ketamine/xylazine mixture (100/10 mg/kg, i.p.) and then given bilateral microinjections of 0.8 µl/side of lentivirus solution at a rate of 0.1 µl/min. The following stereotaxic coordinates were used for viral delivery: +1.9 mm (anterior/posterior), +0.75 (lateral), −2.75 (dorsal/ventral) at an angle of 15° from the midline (relative to bregma). Animals were left to recover for 4–5 weeks before behavioral testing. The correct placement of the injection site was verified histologically at the end of the experiments (Fig. S5A).

*Gpr56 agonist*: To test the antidepressant-like effect of Gpr56 agonists, we bilaterally infused synthetic peptides (P7 and P19) as well as control or an inactive modified peptide (P19 Y ->N: "TNFAVLMQLSPALVPAELL-NH2") previously described[12], both in the PFC and in the NAcc of mice. Mice were anesthetized with a ketamine/xylazine mixture (100/10 mg/kg, i.p.) and stereotaxically implanted with 12 mm long cannulae in the left and right PrL Area (anterior (AP) + 1.9 from the bregma; lateral (ML) +/− 0.5; ventral (DV) −1.3) or in the left and right nucleus accumbens (NAcc) (AP + 1,6; ML +/− 0.7; DV −3.3). Animals were left to recover for at least 7 days. On the test day, infusion needles (30 Gauge) were inserted into the canulae (needles were 13 mm long i.e. ending 1 mm deeper than the guide canulae) and mice were locally infused with a pump (UNIVENTOR), at a rate of 0.5 µl/min, with P7 (0.5 mM, 1 mM or 2 mM) or vehicle (vehicle: 80% saline + 10% DMSO + 10% Cremophor), or with P19 (1 mM) or its inactive control peptide P19YN (1 mM). The needles were left in place for another 2 min to ensure compound diffusion. Mice were subsequently placed in their cage until the TST session (30 min after infusion).

*Behavioral studies*: Behavioral testing was performed using 7–22 animals per group.

Tail suspension test (TST): Immobility was measured in the TST as previously described[35]. Mice were individually tail-suspended by using a paper adhesive tape that was placed 1 cm from the tip of the tail, in such manner as to down rate the probability of the mice reaching their tail. Immobility time (seconds) was measured during a 6 min test period. In case of the mouse catching its tale the measure was discarded. We tested 8–15 mice/group.

Forced swimming test (FST): The forced swimming test was conducted in clear plastic cylinders (diameter 20 cm; height 25 cm) filled with 6 cm of water (22–25 °C) for 6 min. The duration of immobility was measured manually during the last 4 min of the 6 min trial. A mouse was regarded as immobile when floating motionless or making only those movements necessary to keep its head above the water. We tested 11–22 mice/group.

Locomotor activity: Horizontal activity (ambulations) was assessed in transparent activity cages (20 × 15 × 25 cm), with automatic monitoring of photocell beam breaks (Imetronic, France). Locomotor activity (ambulations defined as breaking two consecutive beams) was recorded for a 1-h period and we conducted analyses between groups, both for the first 6 min and the total duration of the test. We tested 7–21 mice/group.

O-maze: The O-maze consisted of a white circular path (runway width 5.5 cm, Ø = 46 cm) with two opposing compartments protected by walls (height = 10 cm) and two open sectors of equal size. The maze was elevated 40 cm above the ground and illuminated from the top with white light (50 Lux). At the start of the testing session, mice were placed at the end of one of the two closed compartments. The test was recorded with a camera for 5 min. We tested 10–18 mice/group.

Sucrose consumption: For the sucrose preference test mice were first habituated to drink from two graduated pipettes: one filled with water, and the other with sucrose solution for 3 days. The side of the sucrose pipette was alternated each day. On day 4 and after an overnight (15 h) deprivation of water, the two pipettes were presented again; however, one was filled with water and the other with 4% sucrose. The water and sucrose solution consumed over a 3-h period were measured. The sucrose preference index is defined as (sucrose consumed)/(sucrose consumed + water consumed). We tested 11–16 mice/group.

Cognitive testing: To test the impact of mouse PFC *Gpr56* inhibition on cognition, we used an attentional set shifting test (ASST) for mice as previously described[10,36]. The extra-dimensional shift task of the ASST has been associated with medial-frontal lesion and is considered as a measure of executive functioning associated with PFC in primate and rodent animal models.

*Measurement of Gpr56 expression:* RNA extractions: Brain punches were made with a Rodent Brain Matrice, ASI Instruments RBM-2000C, and a Harris Uni-Core, Hole 1.0 mm biopsy tool. Regions were punched according to[37]. Total RNA from tissue punches was obtained using Trizol® reagent (Invitrogen, France) and 1 μg was reverse transcribed with random primers from Biolabs (Beverly, MA) and Reverse Transcriptase MLV-RT from Fisher scientific (France), according to manufacturers' instructions.

Real-time quantitative polymerase chain reaction (qRT-PCR): qRT-PCR was performed to assess *Gpr56* expression using SYBR green (ABgene, France) on an ABI PRISM 7000. Each reaction was performed in triplicate and the mean of at least three independent experiments was calculated. All results were normalized to 26S or *Gapdh* and calculated using the relative ($2^{-\Delta\Delta Ct}$) quantification method. The primer sequences used in real time PCR are: 26S (FW: AGGAGAAACAACGG TCGTGCCAAAA, REV: GCGCAAGCAGGTCTGAATCGTG), and GPR56 (FW: TCCAGGCATACTCGCTGTTGCT, REV: CTTCTCACCCAGGACTTGGCTA).

**Cell experiments**. *Cell Culture:* Human neuroblastoma cells (SK-N-AS, ATCC CRL-2137) were cultured in Dulbecco's Modified Eagle Medium (DMEM) supplemented with 10% FBS, 1% non-essential amino acids, 100 U/ml penicillin and 100 μg/ml streptomycin (Invitrogen) in a 5% $CO_2$ humidified incubator at 37 °C. Cells were treated with 25 μM of peptide (P7, P19, P19Y N) or vehicle (DMSO) for 24 h then collected in TRI reagent. RNA was extracted using the DirectZol kit with DNase treatment (Zymo). Three experiments were performed in triplicate. For sequencing, we pooled the triplicates from each experiment.

*RNA sequencing:* All libraries were prepared using the NEB mRNA stranded protocol following the manufacturer's instructions. Samples were sequenced at the McGill University and Genome Quebec Innovation Centre (Montreal, Canada) using the Illumina HiSeq4000 with 100nt paired-end reads. Based on the number of reads, their length and the estimated human exome size being around 3 Mb, the average sequencing depth across all samples is 115×.

FASTX Toolkit (v0.0.14) and Trimmomatic (v0.36) were respectively used for quality and adapter trimming. TopHat (v2.1.1), using Bowtie2 (v2.2.9) was used to align the cleaned reads to the reference genome (GRCh38, https://www.ncbi. nlm.nih.gov/assembly/GCF_000001405.39). Reads that lost their mates through the cleaning process were aligned independently from the reads that still had pairs. Quantification on each gene's expression was estimated using HTSeq-count and a reference transcript annotation from ENSEMBL (v77). Counts for the paired and orphaned reads for each sample were added to each other. Genes counts for each sequenced library were normalized using DESeq2's median ratio normalization method[38].

To facilitate downstream analyses, we chose to correct our normalized counts for the effect of potential covariates using limma's removeBatchEffect function[39]. We specifically regressed out the effects of a possible batch effect associated with the cell culture as well the expected heterogeneity associated with the use of the two different peptides and their respective controls. Our analysis demonstrated that GPR56 was expressed in each cell line.

*Gene set enrichment analysis:* To functionally characterize the gene expression variation associated with GPR56 agonist treatment, we used gene set enrichment analysis[15]. Based on the largest differences in expression between cells receiving agonist or not, GSEA allowed us to calculate enrichment for predefined gene sets related to functional pathways based on enrichment scores and *p*-values, as well as Familywise-error rate FWER $p < 0.20$[40], adjusted for gene set size and multiple hypotheses testing. We used gene sets previously described[41].

**Statistical analysis**. Data collection was conducted using Excel 2013 or Epidata V3.1.

For the discovery cohort, data were expressed as proportions and frequency for categorical variables or means and standard deviations (SD) for continuous variables. Differences between groups were compared using Chi-square tests for categorical data or two-sided *t*-tests for continuous variables. Repeated measures were analyzed using paired *t*-tests. Moreover, group by time interactions were evaluated using GLM for repeated measures. If necessary, potential confounding factors according to univariate analyses and/or current knowledge were also added in GLM. Correlations between continuous variables were conducted using Pearson correlation coefficient calculation and partial correlation analysis to include potential confounding factors.

For microarray expression data analysis in the discovery cohort, we first used two-class paired significant analysis of microarray (SAM)[42] to determine differential gene expression before and after treatment in the duloxetine responders group with MultiExperiment Viewer 4 (MeV4, TM4 software suite). False discovery rate (FDR) threshold was set at 1% and *q*-values were computed. To identify specific probes which were differentially expressed between the two time points in the duloxetine responders group, we also performed two-class paired *t*-test comparisons between both times in the duloxetine non-responders group, the placebo responders group, and the placebo non-responders group. To confirm

specificity of gene expression variation across time in the duloxetine responders group, we built a GLM for repeated measures including all available samples to identify the effect of time, group (placebo or duloxetine) and response, as well as the interaction between them. To control for potential confounding factors, we also included age, gender and BMI.

For the replication cohorts (Montréal cohort and Marseille cohort), we performed a two-paired *t*-test to identify gene expression variation across time in responders, non-responders and healthy controls. To confirm specificity in the responders, we built a GLM for repeated measures as described above. For the Marseille cohort, we also conducted a Linear Mixed Model to compare *GPR56* mRNA between groups across the four visits.

For the post-mortem cohort, we used a Mann–Whitney test, according to the skewness of the data, to compare the level of expression of *GPR56* between healthy controls and individuals with depression who died by suicide. We log2-transformed raw data to achieve a normal distribution, in order to conduct a GLM to control for potential confounding factors.

For animal experiments, to compare behavior measures and gene expression levels, we used t-tests, one-way or two-way ANOVAs and adapted post hoc test. Correlation analyses were conducted using Pearson's coefficient calculation.

All statistical analyses were conducted using SPSS V21 and GraphPad Prism 5 and *p*-value < 0.05 was considered as significant, except where noted.

**Reporting summary**. Further information on experimental design is available in the Nature Research Reporting Summary linked to this paper.

## Data availability
The source data underlying Figs. 1a–c, 2b, c, 3b–e, 4a, b, Supplementary Figs. 1–12 and Supplementary Tables 1 and 2 are provided as a Source Data file. Microarray data obtained in this study are available under accession code GSE146446.

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

## Acknowledgements
We are grateful to all patients for their involvement in this research. This manuscript contains data from four registered clinical trials: www.ClinicalTrials.gov NCT00635219, NCT00599911, NCT01140906, and NCT02209142. Clinical trials NCT00635219, NCT00599911, and NCT01140906 were sponsored by Lundbeck, and samples were provided as a donation to the Canadian Biomarker Integration Network in Depression (CAN-BIND) program. This research was conducted with the support of CAN-BIND, an Integrated Discovery Program, with funding from the Ontario Brain Institute, an independent non-profit corporation, funded partially by the Ontario government. RB received a grant from Fondamental Foundation (Post-doctoral fellowship). RB and EI were supported by the Agence Nationale de la Recherche (ANR-13-SAMA-0002). VG holds a Labex-Biopsy Fellowship. ETT is supported by Fondation de Recherche sur le Cerveau and Fondation de France. ETT is a past recipient of the Bodossakis Foundation Young Scientist Award. RB, ECI, ETT and GT were supported by ERA-NET NEURON (Grant ANTARES). GT holds a Canada Research Chair (Tier 1) and a NARSAD Distinguished Investigator Award. He is supported by grants from the Canadian Institute of Health Research (CIHR) (FDN148374 and EGM141899), and by the *Fonds de recherche du Québec – Santé* (FRQS) through the Quebec Network on Suicide, Mood Disorders and Related Disorders. We would like to thank ML Niepon (Institute of Vision, Paris) for slide scanning for the FISH experiments.

## Author contributions
R.B., E.T.T., S.D.H., G.G.T., and G.T. designed the study. R.B., V.G., L.M.F., E.C.I., E.G., E.T.T., and G.T. wrote the manuscript. R.B., E.C.I., P.C., S.R.D., M.B., E.C., S.D.H., and G.T. conducted the clinical studies (inclusion, evaluation of subjects and analyses of clinical data). R.B., E.C.I., and J.P.L. performed the sample processing from clinical studies. R.B., J.P.L., R.L., and E.C.I. performed the gene expression studies in the clinical cohorts. R.B., J.P.L., C.N., and N.M. performed the experiments on post-mortem brain samples. V.C. and J.G. performed and analysed the animal experiments. L.M.F. and R.L. performed the cell experiments. G.G.T. produced agonist peptides. S.D. performed FISH analyses. R.B., V.G., L.M.F., E.T.T., and J.F.T. performed the statistical analyses. C.N.S., S.R., J.A.F., and B.G. contributed to study design and writing the manuscript.

## Competing interests
The authors declare no competing interests.
