## [Peer Review File · Nature Communications]

Reviewers' comments:

Reviewer #1 (Remarks to the Author):

In this manuscript the authors examine GPR56, which is a G-protein coupled receptor, as a correlate of antidepressant efficacy and depression. While the manuscript presents interesting data that may be beneficial for the development of potential biomarkers to evaluate antidepressant responses and possibly better diagnose depression, the absence of information on the GPR56 signaling pathway and its impact on neurotransmission or signaling mechanisms linked to depression/stress severely limits the significance of the study. Overall, the authors perform experiments to establish a correlation between depression-like behaviors and antidepressant responses with GPR56 levels. That said, the manuscript does not present any information on the ligands that interact with GPR56 and how activation of these receptors in blood or in the CNS impacts neuronal function and downstream neural circuits. Moreover, although GPR56 levels in blood may show correlations with depression-related behaviors and antidepressant efficacy, without a better understanding of the physiology of GPR56 signaling in blood and its relationship to the pathophysiology of depression, or mechanisms underlying antidepressant action, this information is difficult to link causally to any of the processes investigated. Taken together, the limited neurobiological information provided about GPR56 in this manuscript substantially reduces the significance and impact of the study.

Reviewer #2 (Remarks to the Author):

This is truly an interesting paper, but it is a pity that there is a lack of any mechanistic data. The authors should use tissues available from their knockdown/over-expression animal models to test at least some of the most likely biological phenotypes related to depression and/or antidepressant response, at a minimum the levels of stress/neuroplasticity-related molecules in available brain samples.

Reviewer #3 (Remarks to the Author):

This study identifies Gpr56 as a regulated gene in blood of depressed patients that is reversed by antidepressant treatment in responders but not in nonresponders, including an initial discovery cohort and several validation cohorts. The results also show that exposure of mice to unpredictable chronic mild stress (UCMS) decreases the expression of Gpr56 in blood and that antidepressant treatment reverses this effect in responders, similar to the effects observed in depressed patients. Moreover, similar regulation of Gpr56 is observed in the PFC, but not other brain regions examined in mice. The impact of altered Gpr56 expression in the PFC is also examined, and the results show that overexpression of Gpr56 in the PFC produces antidepressant actions in the TST and FST, two models of antidepressant response. In addition, shRNA knockdown of Gpr56 in the PFC resulted in depressive like behaviors in the TST and FST, as well as in the sucrose preference test and O-maze anxiety test. Finally, studies of postmortem PFC of suicide depressed subjects shows that levels of Gpr56 are also decreased, providing an important association between Gpr56 and depressive pathophysiology. There are several points to address.

1. In figure, 2B and 2C, why are there different numbers of animals? Were different cohorts used for the blood vs brain analysis?

2. What was the time frame for the ambulations analysis in Figure 6A? A time period similar to the TST should be examined since Figure S7A shows a trend for increased ambulation after Gpr56 overexpression, which could contribute to decreased immobility in the FST.

3. In the results the authors state that "This observation was further strengthened by similar results in the forced swim test (FST) and consistent behavioural effects for Gpr56 downregulation

in the sucrose preference and O-maze tests (Fig. S6B-D and S7B-D).” However, Fig S6 only shows the effects of Gpr5 overexpression, not down regulation. This should be corrected as overexpression of Gpr56 has no effect on sucrose preference or O-maze.

4. The impact of the overexpression studies would be increased if conducted in mice exposed to UCMS to determine if the stress induced deficits could be reversed?
5. The subregion of the PFC that was examined in postmortem brains of depressed suicide should be stated.
6. What is relative abundance of Gpr56 in different brain regions? What is cell types express Gpr56? Additional in situ hybridization or immunohistochemistry studies should be conducted to examine this issue.
7. The impact and significance of the Gpr56 expression and knockdown studies would be increased if this were examined in a cell type specific manner.

Reviewer #1 (Remarks to the Author):

In this manuscript the authors examine GPR56, which is a G-protein coupled receptor, as a correlate of antidepressant efficacy and depression. While the manuscript presents interesting data that may be beneficial for the development of potential biomarkers to evaluate antidepressant responses and possibly better diagnose depression, the absence of information on the GPR56 signaling pathway and its impact on neurotransmission or signaling mechanisms linked to depression/stress severely limits the significance of the study.

Overall, the authors perform experiments to establish a correlation between depression-like behaviors and antidepressant responses with GPR56 levels. That said, the manuscript does not present any information on the ligands that interact with GPR56 and how activation of these receptors in blood or in the CNS impacts neuronal function and downstream neural circuits.

Moreover, although GPR56 levels in blood may show correlations with depression-related behaviors and antidepressant efficacy, without a better understanding of the physiology of GPR56 signaling in blood and its relationship to the pathophysiology of depression, or mechanisms underlying antidepressant action, this information is difficult to link causally to any of the processes investigated. Taken together, the limited neurobiological information provided about GPR56 in this manuscript substantially reduces the significance and impact of the study.

We thank REV1 for the important comments on our manuscript, which prompted us to conduct additional work to better understand the mechanisms whereby GPR56 may be associated with antidepressant action, including exploration of GPR56 signaling pathways and how it may impact neuronal function. More precisely, (1) we pharmacologically activated the GPR56 receptor using agonist peptides, and investigated the resulting behavioral effects in mice and (2) we defined, using a cellular model, the GPR56 signaling cascade. Finally, (3) we expanded our Discussion to include current knowledge regarding ligands of GPR56.

1) Behavioral experiments using GPR56 agonists

We conducted new experiments in mice using agonists of GPR56 that have been developed and characterized by Dr. Tall's laboratory and reported elsewhere ¹. Using these synthetic peptides (P7 and P19) as well as an inactive peptide (P19 Y -> N: "TNFAVLMQLSPALVPAELL-NH2" also called P19YN), we tested the antidepressant effects of GPR56 activation on depressive-like behaviors using the tail suspension test, a commonly used test of antidepressant effects ².

First, we bilaterally infused the GPR56 agonist P7 peptide, in escalating doses, into the mouse prefrontal cortex (PFC) because our findings indicated that GPR56 was dysregulated in the prefrontal cortex both of individuals with major depressive disorder as well as in mice who displayed depressive-like behaviors. As shown in the Figure below (Fig. 3E in the revised version of the manuscript), we found a significant antidepressant effect of both the 1mM and 2mM doses of P7. As 1mM of agonist produced the most significant antidepressant effects, we repeated this experiment with the same dose of P19. As shown in Supplemental Figure (Fig. S9), bilateral infusion of P19 into the PFC also had antidepressant effects.

Secondly, to confirm the specificity of GPR56 activation in the PFC for antidepressant effects, we assessed the effects of the peptides when infused into the NAcc. We found no differences in the TST between groups, as shown below (Fig. S10). This finding was consistent with the data from the UCMS experiments where GPR56 changes were specific to the PFC.

Finally, to ensure that the effects of the GPR56 agonists were due to their antidepressant effects, we assessed the locomotor activity of the mice, and found no differences between groups (see below and in Fig. S11).

These new data complement the experiments reported in the original manuscript and provide evidence that activation of GPR56 through pharmacological manipulation by GPR56-specific ligands has antidepressant-like effects, specifically in PFC. These experiments are consistent with the findings reported in the original manuscript, and further support a role of GPR56 in depressive-like behaviors and antidepressant response. Furthermore, they indicate that GPR56 may represent a new molecular target for treatment of major depressive disorder.

Fig. 3E: GPR56 agonist P7 has antidepressant-like effects. GPR56 agonist P7 peptide infused bilaterally with escalating doses in PFC decreases immobility time and demonstrated antidepressant-like effects, ANOVA $F(1,4)=4.88$ $p=0.008$. In comparison to vehicle ($N=13$), both 1 mM ($N=7$) and 2 mM ($N=4$) doses demonstrated a significant decrease of immobility time ($p<0.01$ and $p=0.01$ respectively, post-hoc test). Bars represent mean. ** $p<0.01$.

Fig. S9: GPR56 agonist P19 peptide infused bilaterally in PFC (1 mM) decreased immobility time and demonstrated antidepressant-like effects ($t=3.775$, $p=0.006$).

Fig. S10: Effect of infusion of peptide agonists into the NAcc on depressive-like behavior.

Fig. S11: Effect of peptide agonists infused into the PFC on basic locomotion.

The following has been added to the Results section:

Effects of Gpr56 agonist on mouse behavior:

Following activation of the GPR56 receptor by its ligands, the extracellular and transmembrane domains of GPR56 dissociate to reveal a tethered-peptide-agonist³. Based on this mechanism, synthetic peptides (i.e. P7 “TYFAVLM-NH₂” and P19 “TYFAVLMQLSPALVPAELL-NH₂”), comprising the specific portion of tethered-peptide-agonist, have been generated and demonstrate GPR56 agonist properties^{1,3}. We bilaterally infused

these peptides and their inactive controls in the mouse PFC to explore the behavioral effects of GPR56 activation. Parallel to our results with Gpr56 overexpression, behavioral analyses showed that GPR56 agonists produced antidepressant-like effects in unstressed mice, as seen by decreased immobility in the tail suspension test (TST) for P7 with a dose-response profile (Fig. 3E, ANOVA $F(1,4)=4.88$ $p=0.008$) and for P19 (Fig. S9). Interestingly, we confirmed the specificity of antidepressant-like effects of GPR56 in the PFC by using the same peptides infused in the NAcc, which produced no behavioral effects (Fig. S10). Finally, the antidepressant-like effect of the GPR56 agonists was not explained by basic locomotion differences, as we did not find any differences in ambulations across time between active peptides and their controls (Fig. S11). These data provide evidence that activation of GPR56 through pharmacological manipulation by GPR56-specific ligands has antidepressant-like effects, specifically in the PFC. These experiments further support a role of GPR56 in depressive-like behaviors and antidepressant response. Furthermore, they indicate that GPR56 may represent a new molecular target for treatment of MDD.

The following has been added to the Methods:

“To test the antidepressant-like effect of GPR56 agonists, we bilaterally infused synthetic peptides (P7 and P19) as well as control or an inactive modified peptide (P19 Y → N: “TNFAVLMQLSPALVPAELL-NH₂”) previously described¹, both in the PFC and in the NAcc of mice. Mice were anesthetized with a ketamine/xylazine mixture (100/10 mg/kg, i.p.) and stereotaxically implanted with 12mm long canulae in the left and right PrL Area (anterior (AP) +1.9 from the bregma; lateral (ML) +/-0.5; ventral (DV) -1.3) or in the left and right nucleus accumbens (NAcc) (AP +1,6; ML +/- 0.7; DV - 3,3). Animals were left to recover for at least 7 days. On the test day, infusion needles (30 Gauge) were inserted into the canulae (needles were 13mm long i.e. ending 1mm deeper than the guide canulae) and mice were locally infused with a pump (UNIVENTOR), at a rate of 0.5µl/min, with P7 (0.5 mM, 1 mM or 2 mM) or vehicle (vehicle: 80% saline + 10% DMSO + 10% Cremophor), or with P19 (1 mM) or its inactive control peptide P19YN (1 mM). The needles were left in place for another 2 minutes to ensure compound diffusion. Mice were subsequently placed in their cage until the TST session (30 minutes after infusion).”

2) Description of biological processes associated with GPR56 activation

In order to gain a more complete understanding of downstream signaling processes initiated by GPR56 activation, we investigated the transcriptional consequences of treatment with the two GPR56 agonists described above. These experiments were performed in vitro, using a human neuroblastoma cell line, which was treated with the agonists for 24 hours then examined using RNA sequencing. We used these cells because they are derived from neural cells, express GPR56 receptors, and express genes from several important pathways that have been associated with antidepressant response^{4,5}.

To functionally characterize the gene expression variation associated with the GPR56 agonists, we used Gene Set Enrichment Analysis⁶. We determined Familywise-error rate FWER $p<0.20$ as statistical threshold⁷.

6,568 gene sets with sizes between 15 to 500 genes were included in the analysis after gene set size filtering. Among them, we identified significant enrichment of 9 different gene sets (FWER<0.20, Table S4 below). Interestingly, AKT, GSK3 and EIF4 pathways demonstrated the highest normalized enrichment scores and lower FWER p-value. These pathways were upregulated in cells treated with the agonists, in comparison to control conditions. These pathways are highly related and have been described as downstream biological mechanisms involved in depression and antidepressant action of several different drugs, including SSRIs and ketamine^{8,9,10,11}. As a consequence, our results suggest

that GPR56 agonists may have antidepressant effects through pathways that are similar to those activated by established antidepressants.

TABLE S4: Gene Set Enrichment Analysis described several gene sets associated with the transcriptional signature of GPR56 agonists in neuroblastoma cells (FWER p-value < 0.20).

	SOURCE	PATHWAY SIZE	Enrichment Score	Normalized Enrichment Score	p-val	FWER p-val
Gene Set up-regulated by GPR56 AGONIST						
AKT PATHWAY	BIOCARTA	19	-0.60	-2.38	<0.001	0.02
EIF4 PATHWAY	BIOCARTA	23	-0.57	-2.14	<0.001	0.08
GSK3 PATHWAY	BIOCARTA	25	-0.60	-2.12	0.01	0.09
VIRAL GENOME REPLICATION	GO Biological Process GO:0019079	23	-0.47	-2.10	<0.001	0.11
TFF PATHWAY	BIOCARTA	19	-0.57	-2.06	0.00	0.15
Gene Set down-regulated by GPR56 AGONIST						
POLY(A)+ MRNA EXPORT FROM NUCLEUS	GO Biological Process GO:0016973	15	0.72	2.43	<0.001	0.01
ESTABLISHMENT OF SPINDLE LOCALIZATION	GO Biological Process GO:0051293	44	0.49	2.12	0.00	0.10
PROTEIN K48-LINKED UBIQUITINATION	GO Biological Process GO:0070936	52	0.43	2.09	0.02	0.12
POSITIVE REGULATION OF LAMELLIPODIUM ORGANIZATION	GO Biological Process GO:1902745	26	0.61	2.05	<0.001	0.17

The following has been added to the Results:

GPR56 agonists up-regulate AKT/GSK3/EIF4 pathways in neuroblastoma cells

In order to gain a more complete understanding of downstream signaling processes initiated by GPR56 activation, we investigated the transcriptional consequences of treatment with the two GPR56 agonists described above. These experiments were performed *in vitro*, using a human neuroblastoma cell line, which was treated with the agonist peptides for 24 hours then examined using RNA sequencing. We used these cells because they are derived from neural cells, express GPR56 receptors, and express genes from several important pathways that have been associated with antidepressant response, including the serotonin signaling pathway^{4,5}.

To functionally characterize the gene expression variation associated with agonist-induced activation of GPR56, we used Gene Set Enrichment Analysis⁶. 6,568 gene sets with sizes between 15 to 500 genes were included in the analysis after gene set size filtering. Among them, we identified significant enrichment of 9 gene sets (FWER<0.20, Table S4). Interestingly, AKT, GSK3 and EIF4 pathways demonstrated the highest normalized enrichment scores and lowest FWER p-values for upregulated gene sets. These pathways were upregulated in cells treated with the agonists, in comparison to control conditions. These pathways are highly related and have been described as downstream biological mechanisms involved in depression and antidepressant action of several different drugs, including SSRIs and ketamine^{8, 9, 10, 11}. As a consequence, our results suggest that GPR56

agonists may have antidepressant effects through pathways that are similar to those activated by established antidepressants.

The following has been added to the Discussion:

Using two agonist peptides, we confirmed in mice that activation of GPR56 in the PFC is associated with behavioral responses that are commonly associated with antidepressant treatment. Moreover, based on cellular experiment and RNA sequencing, we found that GPR56 agonists up-regulated AKT-GSK3-EIF4 pathways, downstream biological mechanisms associated with depression and antidepressants action^{8, 9, 10, 11}. Overall, our results suggest that GPR56 is a potential new target for development of new antidepressant drugs.

The following has been added to the Methods:

Cell experiments

Cell Culture: Human neuroblastoma cells (SK-N-AS, ATCC CRL-2137) were cultured in Dulbecco's Modified Eagle Medium (DMEM) supplemented with 10% FBS, 1% non-essential amino acids, 100 U/ml penicillin and 100 µg/ml streptomycin (Invitrogen) in a 5% CO₂ humidified incubator at 37°C. Cells were treated with 25µM of peptide (P7, P19, P19Y N) or vehicle (DMSO) for 24 hours then collected in TRI reagent. RNA was extracted using the DirectZol kit with DNase treatment (Zymo). Three experiments were performed in triplicate. For sequencing, we pooled the triplicates from each experiment.

RNA sequencing: All libraries were prepared using the NEB mRNA stranded protocol following the manufacturer's instructions. Samples were sequenced at the McGill University and Genome Quebec Innovation Centre (Montreal, Canada) using the Illumina HiSeq4000 with 100nt paired-end reads. Based on the number of reads, their length and the estimated human exome size being around 3Mb, the average sequencing depth across all samples is 115X.

FASTXToolkit and Trimmomatic were respectively used for quality and adapter trimming. Tophat2, using bowtie2 was used to align the cleaned reads to the reference genome (GRCh38). Reads that lost their mates through the cleaning process were aligned independently from the reads that still had pairs. Quantification on each gene's expression was estimated using HTSeq-count and a reference transcript annotation from ENSEMBL. Counts for the paired and orphaned reads for each sample were added to each other. Genes counts for each sequenced library were normalized using DESeq2's median ratio normalization method¹².

To facilitate downstream analyses, we chose to correct our normalized counts for the effect of potential covariates using limma's removeBatchEffect function¹³. We specifically regressed out the effects of a possible batch effect associated with the cell culture as well the expected heterogeneity associated with the use of the two different peptides and their respective controls. Our analysis demonstrated that *GPR56* was expressed in each cell line.

Gene Set Enrichment Analysis: To functionally characterize the gene expression variation associated with GPR56 agonist treatment, we used Gene Set Enrichment Analysis⁶. Based on the largest differences in expression between cells receiving agonist or not, GSEA allowed us to calculate enrichment for predefined gene sets related to functional pathways based on enrichment scores and p-values, as well as Familywise-error rate FWER p<0.20⁷, adjusted for gene set size and multiple hypotheses testing. We used gene sets previously described¹⁴.

3) GPR56 ligands

GPR56 ligands consist of two general subtypes: 1) proteins from the surface of neighboring cells, and 2) extracellular matrix proteins. Known extracellular ligands of GPR56 include collagen III, transglutaminase 2, and heparin^{15, 16, 17}. Following activation of the GPR56 receptor by its ligands, the extracellular and transmembrane domains dissociate to reveal a tethered-peptide-agonist³.

We added this information in the revised version of the manuscript in the Discussion section.

“GPR56 is involved in a number of biological functions relevant to the pathophysiology of depression, including neurogenesis, oligodendrocyte development and progenitor cell migration in brain, as well as myelin repair^{15, 18, 19, 20}, in parallel to its important role in immune cell functioning^{21, 22, 23}. GPR56 ligands comprise two general subtypes: 1) proteins from the surface of neighboring cells, and 2) extracellular matrix proteins. Known extracellular ligands of GPR56 include collagen III, transglutaminase 2, and heparin^{15, 16, 17}. Following activation of the GPR56 receptor by its ligands, the extracellular and transmembrane domains dissociate to reveal a tethered-peptide-agonist³. To date, it remains unclear which ligand could be related to depressive behavior and antidepressant effects of GPR56 in the PFC. Moreover, GPCRs are particularly appealing drug targets.”

Reviewer #2 (Remarks to the Author):

This is truly an interesting paper, but it is a pity that there is a lack of any mechanistic data. The authors should use tissues available from their knockdown/over-expression animal models to test at least some of the most likely biological phenotypes related to depression and/or antidepressant response, at a minimum the levels of stress/neuroplasticity-related molecules in available brain samples.

We thank the Reviewer for the positive comments. In order to address the Reviewer's comment on mechanisms whereby GPR56 may have antidepressant effects, and to address a similar request from Reviewer #1, we now provide data obtained with two GPR56 agonist peptides, which offer precise information on of the behavioral and molecular consequences of GPR56 activation.

More precisely, (1) we pharmacologically activated the GPR56 receptor using an agonist peptide, and investigated the resulting behavioral effects in mice and (2) we defined, using a cellular model, the GPR56 signaling cascade.

1) Behavioral experiments using GPR56 agonists

We conducted new experiments in mice using agonists of GPR56 that have been developed and characterized by Dr. Tall's laboratory and reported elsewhere ¹. Using these synthetic peptides (P7 and P19) as well as an inactive peptide (P19 Y -> N: "TNFAVLMQLSPALVPAELL-NH2" also called P19YN), we tested the antidepressant effects of GPR56 activation on depressive-like behaviors using the tail suspension test, a commonly used test of antidepressant effects ².

First, we bilaterally infused the GPR56 agonist P7 peptide, in escalating doses, into the mouse prefrontal cortex (PFC) because our findings indicated that GPR56 was dysregulated in the prefrontal cortex both of individuals with major depressive disorder as well as in mice who displayed depressive-like behaviors. As shown in the Figure below (Fig. 3E in the revised version of the manuscript), we found a significant antidepressant effect of both the 1mM and 2mM doses of P7. As 1mM of agonist produced the most significant antidepressant effects, we repeated this experiment with the same dose of P19. As shown in Supplemental Figure (Fig. S9), bilateral infusion of P19 into the PFC also had antidepressant effects.

Secondly, to confirm the specificity of GPR56 activation in the PFC for antidepressant effects, we assessed the effects of the peptides when infused into the NAcc. We found no differences in the TST between groups, as shown below (Fig. S10). This finding was consistent with the data from the UCMS experiments where GPR56 changes were specific to the PFC.

Finally, to ensure that the effects of the GPR56 agonists were due to their antidepressant effects, we assessed the locomotor activity of the mice, and found no differences between groups (see below and in Fig. S11).

These new data complement the experiments reported in the original manuscript and provide evidence that activation of GPR56 through pharmacological manipulation by GPR56-specific ligands has antidepressant-like effects, specifically in PFC. These experiments are consistent with the findings reported in the original manuscript, and further support a role of GPR56 in depressive-like behaviors and antidepressant response. Furthermore, they indicate that GPR56 may represent a new molecular target for treatment of major depressive disorder.

Fig. 3E: GPR56 agonist P7 has antidepressant-like effects. GPR56 agonist P7 peptide infused bilaterally with escalating doses in PFC decreases immobility time and demonstrated antidepressant-like effects, ANOVA $F(1,4)=4.88$ $p=0.008$. In comparison to vehicle ($N=13$), both 1 mM ($N=7$) and 2 mM ($N=4$) doses demonstrated a significant decrease of immobility time ($p<0.01$ and $p=0.01$ respectively, post-hoc test). Bars represent mean. ** $p<0.01$.

Fig. S9: GPR56 agonist P19 peptide infused bilaterally in PFC (1 mM) decreased immobility time and demonstrated antidepressant-like effects ($t=3.775$, $p=0.006$).

Fig. S10: Effect of infusion of peptide agonists into the NAcc on depressive-like behavior.

Fig. S11: Effect of peptide agonists infused into the PFC on basic locomotion.

The following has been added to the Results section:

Effects of Gpr56 agonist on mouse behavior:

Following activation of the GPR56 receptor by its ligands, the extracellular and transmembrane domains of GPR56 dissociate to reveal a tethered-peptide-agonist³. Based on this mechanism, synthetic peptides (i.e. P7 “TYFAVLM-NH2” and P19 “TYFAVLMQLSPALVPAELL-NH2”), comprising the specific portion of tethered-peptide-agonist, have been generated and demonstrate GPR56 agonist properties^{1,3}. We bilaterally infused these peptides and their inactive controls in the mouse PFC to explore the behavioral effects of GPR56 activation. Parallel to our results with Gpr56 overexpression, behavioral analyses showed that GPR56 agonists produced antidepressant-like effects in unstressed mice, as seen by decreased immobility in the tail suspension test (TST) for P7 with a dose-response profile (Fig. 3E, ANOVA $F(1,4)=4.88$ $p=0.008$) and for P19 (Fig. S9). Interestingly, we confirmed the specificity of antidepressant-like effects of GPR56 in the PFC by using the same peptides infused in the NAcc, which produced no behavioral effects (Fig. S10). Finally, the antidepressant-like effect of the GPR56 agonists was not explained by basic locomotion differences, as we did not find any differences in ambulations across time between active peptides and their controls (Fig. S11). These data provide evidence that activation of GPR56 through pharmacological manipulation by GPR56-specific ligands has antidepressant-like effects, specifically in the PFC. These experiments further support a role of GPR56 in depressive-like behaviors and antidepressant response. Furthermore, they indicate that GPR56 may represent a new molecular target for treatment of major depressive disorder.

The following has been added to the Methods:

“To test the antidepressant-like effect of GPR56 agonists, we bilaterally infused synthetic peptides (P7 and P19) as well as control or an inactive modified peptide (P19 Y → N: “TNFAVLMQLSPALVPAELL-NH₂”) previously described¹, both in the PFC and in the NAcc of mice. Mice were anesthetized with a ketamine/xylazine mixture (100/10 mg/kg, i.p.) and stereotaxically implanted with 12mm long canulae in the left and right PrL Area (anterior (AP) +1.9 from the bregma; lateral (ML) +/-0.5; ventral (DV) -1.3) or in the left and right nucleus accumbens (NAcc) (AP +1,6; ML +/- 0.7; DV - 3,3). Animals were left to recover for at least 7 days. On the test day, infusion needles (30 Gauge) were inserted into the canulae (needles were 13mm long i.e. ending 1mm deeper than the guide canulae) and mice were locally infused with a pump (UNIVENTOR), at a rate of 0.5µl/min, with P7 (0.5 mM, 1 mM or 2 mM) or vehicle (vehicle: 80% saline + 10% DMSO + 10% Cremophor), or with P19 (1 mM) or its inactive control peptide P19YN (1 mM). The needles were left in place for another 2 minutes to ensure compound diffusion. Mice were subsequently placed in their cage until the TST session (30 minutes after infusion).”

2) Description of biological processes associated with GPR56 activation

In order to gain a more complete understanding of downstream signaling processes initiated by GPR56 activation, we investigated the transcriptional consequences of treatment with the two GPR56 agonists described above. These experiments were performed *in vitro*, using a human neuroblastoma cell line, which was treated with the agonists for 24 hours then examined using RNA sequencing. We used these cells because they are derived from neural cells, express GPR56 receptors, and express genes from several important pathways that have been associated with antidepressant response^{4,5}.

To functionally characterize the gene expression variation associated with the GPR56 agonists, we used Gene Set Enrichment Analysis⁶. We determined Familywise-error rate FWER $p < 0.20$ as statistical threshold⁷.

6,568 gene sets with sizes between 15 to 500 genes were included in the analysis after gene set size filtering. Among them, we identified significant enrichment of 9 different gene sets (FWER < 0.20, Table S4 below). Interestingly, AKT, GSK3 and EIF4 pathways demonstrated the highest normalized enrichment scores and lower FWER p-value. These pathways were upregulated in cells treated with the agonists, in comparison to control conditions. These pathways are highly related and have been described as downstream biological mechanisms involved in depression and antidepressant action of several different drugs, including SSRIs and ketamine^{8,9,10,11}. As a consequence, our results suggest that GPR56 agonists may have antidepressant effects through pathways that are similar to those activated by established antidepressants.

TABLE S4: Gene Set Enrichment Analysis described several gene sets associated with the transcriptional signature of GPR56 agonists in neuroblastoma cells (FWER p-value < 0.20).

	SOURCE	PATHWAY SIZE	Enrichment Score	Normalized Enrichment Score	p-val	FWER p-val
Gene Set up-regulated by GPR56 AGONIST						
AKT PATHWAY	BIOCARTA	19	-0.60	-2.38	<0.001	0.02
EIF4 PATHWAY	BIOCARTA	23	-0.57	-2.14	<0.001	0.08
GSK3 PATHWAY	BIOCARTA	25	-0.60	-2.12	0.01	0.09

VIRAL GENOME REPLICATION	GO Biological Process GO:0019079	23	-0.47	-2.10	<0.001	0.11
TFF PATHWAY	BIOCARTA	19	-0.57	-2.06	0.00	0.15
Gene Set down-regulated by GPR56 AGONIST						
POLY(A)+ MRNA EXPORT FROM NUCLEUS	GO Biological Process GO:0016973	15	0.72	2.43	<0.001	0.01
ESTABLISHMENT OF SPINDLE LOCALIZATION	GO Biological Process GO:0051293	44	0.49	2.12	0.00	0.10
PROTEIN K48-LINKED UBIQUITINATION	GO Biological Process GO:0070936	52	0.43	2.09	0.02	0.12
POSITIVE REGULATION OF LAMELLIPODIUM ORGANIZATION	GO Biological Process GO:1902745	26	0.61	2.05	<0.001	0.17

The following has been added to the Results:

GPR56 agonists up-regulate AKT/GSK3/EIF4 pathways in neuroblastoma cells

In order to gain a more complete understanding of downstream signaling processes initiated by GPR56 activation, we investigated the transcriptional consequences of treatment with the two GPR56 agonists described above. These experiments were performed *in vitro*, using a human neuroblastoma cell line, which was treated with the agonist peptides for 24 hours then examined using RNA sequencing. We used these cells because they are derived from neural cells, express GPR56 receptors, and express genes from several important pathways that have been associated with antidepressant response, including the serotonin signaling pathway^{4,5}.

To functionally characterize the gene expression variation associated with agonist-induced activation of GPR56, we used Gene Set Enrichment Analysis⁶. 6,568 gene sets with sizes between 15 to 500 genes were included in the analysis after gene set size filtering. Among them, we identified significant enrichment of 9 gene sets (FWER<0.20, Table S4). Interestingly, AKT, GSK3 and EIF4 pathways demonstrated the highest normalized enrichment scores and lowest FWER p-values for upregulated gene sets. These pathways were upregulated in cells treated with the agonists, in comparison to control conditions. These pathways are highly related and have been described as downstream biological mechanisms involved in depression and antidepressant action of several different drugs, including SSRIs and ketamine^{8, 9, 10, 11}. As a consequence, our results suggest that GPR56 agonists may have antidepressant effects through pathways that are similar to those activated by established antidepressants.

The following has been added to the Discussion:

Using two agonist peptides, we confirmed in mice that activation of GPR56 in the PFC is associated with behavioral responses that are commonly associated with antidepressant treatment. Moreover, based on cellular experiment and RNA sequencing, we found that GPR56 agonists up-regulated AKT-GSK3-EIF4 pathways, downstream biological mechanisms associated with depression and antidepressants action^{8, 9, 10, 11}. Overall, our results suggest that GPR56 is a potential new target for development of new antidepressant drugs.

The following has been added to the Methods:

Cell experiments

Cell Culture: Human neuroblastoma cells (SK-N-AS, ATCC CRL-2137) were cultured in Dulbecco's Modified Eagle Medium (DMEM) supplemented with 10% FBS, 1% non-essential amino acids, 100 U/ml penicillin and 100 µg/ml streptomycin (Invitrogen) in a 5% CO₂ humidified incubator at 37°C. Cells were treated with 25µM of peptide (P7, P19, P19Y N) or vehicle (DMSO) for 24 hours then collected in TRI reagent. RNA was extracted using the DirectZol kit with DNase treatment (Zymo). Three experiments were performed in triplicate. For sequencing, we pooled the triplicates from each experiment.

RNA sequencing: All libraries were prepared using the NEB mRNA stranded protocol following the manufacturer's instructions. Samples were sequenced at the McGill University and Genome Quebec Innovation Centre (Montreal, Canada) using the Illumina HiSeq4000 with 100nt paired-end reads. Based on the number of reads, their length and the estimated human exome size being around 3Mb, the average sequencing depth across all samples is 115X.

FASTXToolkit and Trimmomatic were respectively used for quality and adapter trimming. Tophat2, using bowtie2 was used to align the cleaned reads to the reference genome (GRCh38). Reads that lost their mates through the cleaning process were aligned independently from the reads that still had pairs. Quantification on each gene's expression was estimated using HTSeq-count and a reference transcript annotation from ENSEMBL. Counts for the paired and orphaned reads for each sample were added to each other. Genes counts for each sequenced library were normalized using DESeq2's median ratio normalization method¹².

To facilitate downstream analyses, we chose to correct our normalized counts for the effect of potential covariates using limma's removeBatchEffect function¹³. We specifically regressed out the effects of a possible batch effect associated with the cell culture as well the expected heterogeneity associated with the use of the two different peptides and their respective controls. Our analysis demonstrated that *GPR56* was expressed in each cell line.

Gene Set Enrichment Analysis: To functionally characterize the gene expression variation associated with GPR56 agonist treatment, we used Gene Set Enrichment Analysis⁶. Based on the largest differences in expression between cells receiving agonist or not, GSEA allowed us to calculate enrichment for predefined gene sets related to functional pathways based on enrichment scores and p-values, as well as Familywise-error rate FWER $p < 0.20$ ⁷, adjusted for gene set size and multiple hypotheses testing. We used gene sets previously described¹⁴.

Reviewer #3 (Remarks to the Author):

This study identifies Gpr56 as a regulated gene in blood of depressed patients that is reversed by antidepressant treatment in responders but not in nonresponders, including an initial discovery cohort and several validation cohorts. The results also show that exposure of mice to unpredictable chronic mild stress (UCMS) decreases the expression of Gpr56 in blood and that antidepressant treatment reverses this effect in responders, similar to the effects observed in depressed patients. Moreover, similar regulation of Gpr56 is observed in the PFC, but not other brain regions examined in mice. The impact of altered Gpr56 expression in the PFC is also examined, and the results show that overexpression of Gpr56 in the PFC produces antidepressant actions in the TST and FST, two models of antidepressant response. In addition, shRNA knockdown of Gpr56 in the PFC resulted in depressive like behaviors in the TST and FST, as well as in the sucrose preference test and O-maze anxiety test.

Finally, studies of postmortem PFC of suicide depressed subjects shows that levels of Gpr56 are also decreased, providing an important association between Gpr56 and depressive pathophysiology. There are several points to address.

1. In figure, 2B and 2C, why are there different numbers of animals? Were different cohorts used for the blood vs brain analysis?

The same cohort of mice was used for both experiments. Discrepancy between numbers of animals is explained by the fact that RNA from some samples passed quality control for one of the tissues but not the other. We have clarified this in the Figure legend:

“Sample numbers vary between tissues due to removal of poor quality RNA samples from the analyses.”

2. What was the time frame for the ambulations analysis in Figure 6A? A time period similar to the TST should be examined since Figure S7A shows a trend for increased ambulation after Gpr56 overexpression, which could contribute to decreased immobility in the FST.

For all tests (TST, FST, locomotion) in both conditions (virus or control), infused mice were assessed at their basal state. The total ambulations presented in Figure 6A were measured for a period of 60 minutes.

The timeline for the TST and FST is 6 minutes. Therefore, for locomotion, as the reviewer requested, in addition to the total ambulations over 60 minutes, we also present here the first 6 minutes of ambulations, which correspond to the same timeframe of the TST and FST. Thus, locomotion is expressed as ambulations per minute for the first 6 minutes (Figure below).

There is no statistically significant difference in ambulations between both conditions in both experiments.

We did not include these figures in the supplemental but we described this analysis in the Method section as follow:

Methods:

“Locomotor activity: Horizontal activity (ambulations) was assessed in transparent activity cages (20x15x25 cm), with automatic monitoring of photocell beam breaks (Imetronic, France). Locomotor activity (ambulations defined as breaking two consecutive beams) was recorded for a 1-hour period and we conducted analyses between groups, both for the first 6 minutes and the total duration of the test. We tested 7-21 mice/group.”

3. In the results the authors state that “This observation was further strengthened by similar results in the forced swim test (FST) and consistent behavioural effects for Gpr56 downregulation in the sucrose preference and O-maze tests (Fig. S6B-D and S7B-D).” However, Fig S6 only shows the effects of Gpr56 overexpression, not down regulation. This should be corrected as overexpression of Gpr56 has no effect on sucrose preference or O-maze.

We thank the reviewer for pointing this out and agree that this description was unclear. We have revised this as follows:

“This observation was further strengthened by similar results in the forced swim test (FST) for both up- and downregulation of Gpr56 (Fig. S6B and Fig. S7B). We also found behavioural effects for Gpr56 downregulation in the sucrose preference and O-maze tests (Fig. S7C-D).”

4. The impact of the overexpression studies would be increased if conducted in mice exposed to UCMS to determine if the stress induced deficits could be reversed?

This is an interesting point. We agree with the reviewer that exposure to UCMS would allow us to assess whether our results are based on the effects of acute stress (TST, FST) or if they also extend to chronic stress. We have mentioned this limitation in the Discussion section as follows:

“Fourthly, we conducted our animal experiments to test the causal relationship between Gpr56 under- and over-expression, as well as pharmacological testing, only in acute stress paradigms of depressive-like symptoms (i.e. TST, FST).”

5. The subregion of the PFC that was examined in postmortem brains of depressed suicide should be stated.

GPR56 expression was measured in BA44. We have added this information in the legend of the revised Figure 4, as well as added this information to the text as follows:

“Therefore, we next investigated the expression of *GPR56* in the prefrontal cortex (BA44) from individuals who died during an episode of major depressive disorder”

6. What is relative abundance of Gpr56 in different brain regions?

To the best of our knowledge, a detailed analysis of GPR56 expression across the brain has not been previously conducted.

To address the reviewer’s question we performed a gene expression analysis of 16 human brain regions²⁴, and found that GPR56 was expressed in all brain regions (new Supplemental Figure, Fig. S9), with a low level of heterogeneity between brain regions (coefficient of variation = 0.08). In addition, we analyzed Gpr56 expression in the PFC, HV, HD and Nucleus Accumbens of mice and found levels within the same range (qPCR experiments; Fig. 2).

We added this information in the results section of the revised manuscript as follows:

“As suggested by our results in mice (Fig 2), *GPR56* is expressed in all brain regions in human (Fig. S12).”

Fig. S12: Expression of GPR56 across human brain regions based on microarray analysis demonstrated a homogeneous level of expression across brain regions. (Coefficient of Variation=0.08)

What cell types express Gpr56? Additional in situ hybridization or immunohistochemistry studies should be conducted to examine this issue.

To address this question, we first analyzed data from frontal cortex single nucleus RNA-sequencing²⁵, and found that GPR56 was expressed in all cell types.

In addition, as suggested by the reviewer, we performed double labelling fluorescence in situ hybridization of Gpr56 combined with cell-type markers, including astrocyte-specific (Glast), activated astrocyte-specific (GFAP), glutamatergic neuron-specific (Vglut1), GABAergic neuron-specific (Gad1) and oligodendrocyte-specific (MBP) markers. Our results are consistent with the finding that Gpr56 is expressed across different cell types. Here we saw that astrocytes and glutamatergic neurons expressed the highest levels. While these results did not provide clear evidence of double labelling for inhibitory neurons or for oligodendrocytes, they are not inconsistent with the previous findings suggesting that different cell types expressed GPR56.

We have added the following Figure to the manuscript.

Fig. S13. Fluorescence *in situ* hybridization to determine cell-specific expression of Gpr56 in the mouse PFC. GPR56 was co-localized with Glast (astrocyte-specific marker, white arrow in the panel) and Vglut1 (glutamatergic neuron-specific marker, white arrow in the panel).

Additionally, we have added information to the Discussion as follows:

“In the brain, single-cell sequencing data from the frontal cortex indicates that *GPR56* is expressed in all cell types, including glutamatergic neurons, and astrocytes, where relatively higher levels are observed²⁵. These data are consistent, although not fully concordant, with results we observed in the mouse using fluorescence *in situ* hybridization (Fig. S13).”

7. The impact and significance of the Gpr56 expression and knockdown studies would be increased if this were examined in a cell type specific manner.

We agree with the reviewer that this would be an interesting experiment to perform. However, performing these experiments *in vivo* would involve a substantial amount of time and troubleshooting. Instead, we have now included *in vitro* studies in a single cell type. We have expanded the Discussion of the revised manuscript to suggest these experiments for future studies, and have included the single cell type *in vitro* studies, as follows:

“In the future, cell-type specific studies should be conducted in order to elucidate the mechanisms through which GPR56 is involved in depression and antidepressant response.”

In addition, in order to gain a more complete understanding of downstream signaling processes initiated by GPR56 activation, we investigated the transcriptional consequences of treatment with the two GPR56 agonists described above. These experiments were performed *in vitro*, using a human neuroblastoma cell line, which was treated with the agonists for 24 hours then examined using RNA sequencing. We used these cells because they are derived from neural cells, express GPR56

receptors, and express genes from several important pathways that have been associated with antidepressant response^{4,5}.

To functionally characterize the gene expression variation associated with the GPR56 agonists, we used Gene Set Enrichment Analysis⁶. We determined Familywise-error rate FWER $p < 0.20$ as statistical threshold⁷.

6,568 gene sets with sizes between 15 to 500 genes were included in the analysis after gene set size filtering. Among them, we identified significant enrichment of 9 different gene sets (FWER < 0.20 , Table S4 below). Interestingly, AKT, GSK3 and EIF4 pathways demonstrated the highest normalized enrichment scores and lower FWER p-value. These pathways were upregulated in cells treated with the agonists, in comparison to control conditions. These pathways are highly related and have been described as downstream biological mechanisms involved in depression and antidepressant action of several different drugs, including SSRIs and ketamine^{8,9,10,11}. As a consequence, our results suggest that GPR56 agonists may have antidepressant effects through pathways that are similar to those activated by established antidepressants.

TABLE S4: Gene Set Enrichment Analysis described several gene sets associated with the transcriptional signature of GPR56 agonists in neuroblastoma cells (FWER p-value < 0.20).

	SOURCE	PATHWAY SIZE	Enrichment Score	Normalized Enrichment Score	p-val	FWER p-val
Gene Set up-regulated by GPR56 AGONIST						
AKT PATHWAY	BIOCARTA	19	-0.60	-2.38	< 0.001	0.02
EIF4 PATHWAY	BIOCARTA	23	-0.57	-2.14	< 0.001	0.08
GSK3 PATHWAY	BIOCARTA	25	-0.60	-2.12	0.01	0.09
VIRAL GENOME REPLICATION	GO Biological Process GO:0019079	23	-0.47	-2.10	< 0.001	0.11
TFF PATHWAY	BIOCARTA	19	-0.57	-2.06	0.00	0.15
Gene Set down-regulated by GPR56 AGONIST						
POLY(A)+ MRNA EXPORT FROM NUCLEUS	GO Biological Process GO:0016973	15	0.72	2.43	< 0.001	0.01
ESTABLISHMENT OF SPINDLE LOCALIZATION	GO Biological Process GO:0051293	44	0.49	2.12	0.00	0.10
PROTEIN K48-LINKED UBIQUITINATION	GO Biological Process GO:0070936	52	0.43	2.09	0.02	0.12
POSITIVE REGULATION OF LAMELLIPODIUM ORGANIZATION	GO Biological Process GO:1902745	26	0.61	2.05	< 0.001	0.17

The following has been added to the Results:

GPR56 agonists up-regulate AKT/GSK3/EIF4 pathways in neuroblastoma cells

In order to gain a more complete understanding of downstream signaling processes initiated by GPR56 activation, we investigated the transcriptional consequences of treatment with the two GPR56 agonists described above. These experiments were performed *in vitro*, using a human neuroblastoma cell line, which was treated with the agonist peptides for 24 hours then examined using RNA sequencing. We used these cells because they are derived from neural cells, express GPR56 receptors, and express genes from several important pathways

that have been associated with antidepressant response, including the serotonin signaling pathway^{4,5}.

To functionally characterize the gene expression variation associated with agonist-induced activation of GPR56, we used Gene Set Enrichment Analysis⁶. 6,568 gene sets with sizes between 15 to 500 genes were included in the analysis after gene set size filtering. Among them, we identified significant enrichment of 9 gene sets (FWER<0.20, Table S4). Interestingly, AKT, GSK3 and EIF4 pathways demonstrated the highest normalized enrichment scores and lowest FWER p-values for upregulated gene sets. These pathways were upregulated in cells treated with the agonists, in comparison to control conditions. These pathways are highly related and have been described as downstream biological mechanisms involved in depression and antidepressant action of several different drugs, including SSRIs and ketamine^{8,9,10,11}. As a consequence, our results suggest that GPR56 agonists may have antidepressant effects through pathways that are similar to those activated by established antidepressants.

The following has been added to the Discussion:

Using two agonist peptides, we confirmed in mice that activation of GPR56 in the PFC is associated with behavioral responses that are commonly associated with antidepressant treatment. Moreover, based on cellular experiment and RNA sequencing, we found that GPR56 agonists up-regulated AKT-GSK3-EIF4 pathways, downstream biological mechanisms associated with depression and antidepressants action^{8,9,10,11}. Overall, our results suggest that GPR56 is a potential new target for development of new antidepressant drugs.

The following has been added to the Methods:

Cell experiments

Cell Culture: Human neuroblastoma cells (SK-N-AS, ATCC CRL-2137) were cultured in Dulbecco's Modified Eagle Medium (DMEM) supplemented with 10% FBS, 1% non-essential amino acids, 100 U/ml penicillin and 100 µg/ml streptomycin (Invitrogen) in a 5% CO₂ humidified incubator at 37°C. Cells were treated with 25µM of peptide (P7, P19, P19Y N) or vehicle (DMSO) for 24 hours then collected in TRI reagent. RNA was extracted using the DirectZol kit with DNase treatment (Zymo). Three experiments were performed in triplicate. For sequencing, we pooled the triplicates from each experiment.

RNA sequencing: All libraries were prepared using the NEB mRNA stranded protocol following the manufacturer's instructions. Samples were sequenced at the McGill University and Genome Quebec Innovation Centre (Montreal, Canada) using the Illumina HiSeq4000 with 100nt paired-end reads. Based on the number of reads, their length and the estimated human exome size being around 3Mb, the average sequencing depth across all samples is 115X.

FASTXToolkit and Trimmomatic were respectively used for quality and adapter trimming. Tophat2, using bowtie2 was used to align the cleaned reads to the reference genome (GRCh38). Reads that lost their mates through the cleaning process were aligned independently from the reads that still had pairs. Quantification on each gene's expression was estimated using HTSeq-count and a reference transcript annotation from ENSEMBL. Counts for the paired and orphaned reads for each sample were added to each other. Genes counts for each sequenced library were normalized using DESeq2's median ratio normalization method¹².

To facilitate downstream analyses, we chose to correct our normalized counts for the effect of potential covariates using limma's removeBatchEffect function¹³. We specifically

regressed out the effects of a possible batch effect associated with the cell culture as well the expected heterogeneity associated with the use of the two different peptides and their respective controls. Our analysis demonstrated that *GPR56* was expressed in each cell line.

Gene Set Enrichment Analysis: To functionally characterize the gene expression variation associated with GPR56 agonist treatment, we used Gene Set Enrichment Analysis⁶. Based on the largest differences in expression between cells receiving agonist or not, GSEA allowed us to calculate enrichment for predefined gene sets related to functional pathways based on enrichment scores and p-values, as well as Familywise-error rate FWER $p < 0.20$ ⁷, adjusted for gene set size and multiple hypotheses testing. We used gene sets previously described¹⁴.

References

1. Stoveken HM, Larsen SD, Smrcka AV, Tall GG. Gedunin- and Khivorin-Derivatives Are Small-Molecule Partial Agonists for Adhesion G Protein-Coupled Receptors GPR56/ADGRG1 and GPR114/ADGRG5. *Mol Pharmacol* **93**, 477-488 (2018).
2. Svenningsson P, Tzavara ET, Witkin JM, Fienberg AA, Nomikos GG, Greengard P. Involvement of striatal and extrastriatal DARPP-32 in biochemical and behavioral effects of fluoxetine (Prozac). *Proc Natl Acad Sci U S A* **99**, 3182-3187 (2002).
3. Stoveken HM, Hajduczuk AG, Xu L, Tall GG. Adhesion G protein-coupled receptors are activated by exposure of a cryptic tethered agonist. *Proc Natl Acad Sci U S A* **112**, 6194-6199 (2015).
4. Gupta M, *et al.* TSPAN5, ERICH3 and selective serotonin reuptake inhibitors in major depressive disorder: pharmacometabolomics-informed pharmacogenomics. *Mol Psychiatry* **21**, 1717-1725 (2016).
5. Qi XR, Zhao J, Liu J, Fang H, Swaab DF, Zhou JN. Abnormal retinoid and TrkB signaling in the prefrontal cortex in mood disorders. *Cereb Cortex* **25**, 75-83 (2015).
6. Subramanian A, *et al.* Gene set enrichment analysis: a knowledge-based approach for interpreting genome-wide expression profiles. *Proc Natl Acad Sci U S A* **102**, 15545-15550 (2005).
7. Benjamini Y, Hochberg Y. Controlling the False Discovery Rate: A Practical and Powerful Approach to Multiple Testing. *Journal of the Royal Statistical Society Series B (Methodological)* **57**, 289-300 (1995).
8. Zanos P, Gould TD. Mechanisms of ketamine action as an antidepressant. *Mol Psychiatry* **23**, 801-811 (2018).
9. Beaulieu JM. A role for Akt and glycogen synthase kinase-3 as integrators of dopamine and serotonin neurotransmission in mental health. *J Psychiatry Neurosci* **37**, 7-16 (2012).
10. Gould TD, Manji HK. Glycogen synthase kinase-3: a putative molecular target for lithium mimetic drugs. *Neuropsychopharmacology* **30**, 1223-1237 (2005).
11. Aguilar-Valles A, *et al.* Translational control of depression-like behavior via phosphorylation of eukaryotic translation initiation factor 4E. *Nat Commun* **9**, 2459 (2018).
12. Love MI, Huber W, Anders S. Moderated estimation of fold change and dispersion for RNA-seq data with DESeq2. *Genome Biol* **15**, 550 (2014).
13. Ritchie ME, *et al.* limma powers differential expression analyses for RNA-sequencing and microarray studies. *Nucleic Acids Res* **43**, e47 (2015).

14. Merico D, Isserlin R, Stueker O, Emili A, Bader GD. Enrichment map: a network-based method for gene-set enrichment visualization and interpretation. *PLoS One* **5**, e13984 (2010).
15. Giera S, *et al.* Microglial transglutaminase-2 drives myelination and myelin repair via GPR56/ADGRG1 in oligodendrocyte precursor cells. *Elife* **7**, (2018).
16. Luo R, Jeong SJ, Jin Z, Strokes N, Li S, Piao X. G protein-coupled receptor 56 and collagen III, a receptor-ligand pair, regulates cortical development and lamination. *Proc Natl Acad Sci U S A* **108**, 12925-12930 (2011).
17. Chiang NY, *et al.* Heparin interacts with the adhesion GPCR GPR56, reduces receptor shedding, and promotes cell adhesion and motility. *J Cell Sci* **129**, 2156-2169 (2016).
18. Bae BI, *et al.* Evolutionarily dynamic alternative splicing of GPR56 regulates regional cerebral cortical patterning. *Science* **343**, 764-768 (2014).
19. Bai Y, Du L, Shen L, Zhang Y, Zhang L. GPR56 is highly expressed in neural stem cells but downregulated during differentiation. *Neuroreport* **20**, 918-922 (2009).
20. Giera S, *et al.* The adhesion G protein-coupled receptor GPR56 is a cell-autonomous regulator of oligodendrocyte development. *Nat Commun* **6**, 6121 (2015).
21. Peng YM, *et al.* Specific expression of GPR56 by human cytotoxic lymphocytes. *J Leukoc Biol* **90**, 735-740 (2011).
22. Hamann J, *et al.* International Union of Basic and Clinical Pharmacology. XCIV. Adhesion G protein-coupled receptors. *Pharmacol Rev* **67**, 338-367 (2015).
23. Della Chiesa M, *et al.* GPR56 as a novel marker identifying the CD56dull CD16+ NK cell subset both in blood stream and in inflamed peripheral tissues. *Int Immunol* **22**, 91-100 (2010).
24. Sequeira A, *et al.* Global brain gene expression analysis links glutamatergic and GABAergic alterations to suicide and major depression. *PLoS One* **4**, e6585 (2009).
25. Habib N, *et al.* Massively parallel single-nucleus RNA-seq with DroNc-seq. *Nat Methods* **14**, 955-958 (2017).

REVIEWERS' COMMENTS:

Reviewer #1 (Remarks to the Author):

While the authors have spent significant effort to address the questions I raised, a fundamental aspect of my inquiry remains unanswered. In particular, the authors have not produced functional or biochemical data from the animals treated with the activator peptide. I am afraid, I remain unconvinced regarding the robustness of the findings and their mechanistic basis.

Reviewer #2 (Remarks to the Author):

I am happy with the revision

Reviewer #3 (Remarks to the Author):

The authors have responded to the comments and concerns in the revised manuscript including addition of new data.

RESPONSE TO REVIEWERS' COMMENTS:

Reviewer #1 (Remarks to the Author):

While the authors have spent significant effort to address the questions I raised, a fundamental aspect of my inquiry remains unanswered. In particular, the authors have not produced functional or biochemical data from the animals treated with the activator peptide. I am afraid, I remain unconvinced regarding the robustness of the findings and their mechanistic basis.

While we did not directly examine the functional effects of peptide-induced activation of GPR56 *in vivo*, our analyses from the *in vitro* experiments demonstrated that several antidepressant-related pathways were activated following agonist treatment. These results suggest that similar pathways are activated in the mice, and related to their antidepressant response. We have added the following statement to the Discussion.

“Although we did not examine these pathways *in vivo*, it is possible that the upregulation of these pathways explains the antidepressant effects that were observed following agonist treatment.”

Reviewer #2 (Remarks to the Author):

I am happy with the revision

Reviewer #3 (Remarks to the Author):

The authors have responded to the comments and concerns in the revised manuscript including addition of new data.